



# Improving our understanding of wind extremes from Bangladesh tropical cyclones: insights from a high-resolution convection-permitting numerical model

Hamish Steptoe[1] & Theo Economou[2,1]

[1]Met Office, FitzRoy Road, Exeter, EX1 3PB, UK
[2] College of Engineering, Mathematics and Physical Sciences, University of Exeter, UK

*Correspondence to*: Hamish Steptoe (hamish.steptoe@metoffice.gov.uk)

**Abstract.** We use high resolution (4.4km) numerical simulations of tropical cyclones to produce exceedance probability estimates for extreme wind (gust) speeds over Bangladesh. For the first time, we estimate equivalent return periods up to and including a 1-in-200 year event, in a spatially coherent manner over all of Bangladesh, by using generalised additive models. We show that some northern provinces, up to 200 km inland, may experience conditions equal to or exceeding a very severe cyclonic storm event (maximum wind speeds in ≥ 64 knots) with a likelihood equal to coastal regions less than 50 km inland. For the most severe super cyclonic storm events (≥ 120 knots), event exceedance probabilities of 1-in-100 to 1-in-200 events remain limited to the coastlines of southern provinces only. We demonstrate how the Bayesian interpretation of the generalised additive model can facilitate a transparent decision-making framework for tropical cyclone warnings.

## 1 Introduction

Bangladesh is one of the most disaster-prone countries in the world, ranking seventh in the 1999-2018 Long-Term Climate Risk Index (Eckstein et al., 2019). Large portions of the population are exposed to the multiple natural hazards, including those derived from tropical cyclones (TCs), such as high-winds, storm surge and flooding (e.g. Dilley et al., 2005). In the last 30 years, TCs impacting Bangladesh, from the Bay of Bengal (BoB), have been responsible for damages of c.US$5.1 billion and affected 60 million people (Guha-Sapir et al., 2014), with average annual extreme weather event-related losses amounted to 1.8 percent of GDP between 1990 and 2008 (International Monetary Fund, 2019b). The wider North Indian Ocean basin averages 5 cyclone per year (accounting for c.7% of global tropical cyclone activity) (Sahoo and Bhaskaran, 2016); however, there is some indication of a decrease in TC frequency (Alam et al., 2003; Mohapatra et al., 2017; Rao, 2004; Singh et al., 2019) and an increase in cyclone intensity (Balaguru et al., 2014) that is projected to continue under a warming climate (Knutson et al., 2020).

Recently, the IMF (2019b) highlighted the early response Bangladesh is taking to the challenges posed by climate change; however, they also emphasise the importance of insurance mechanisms to enhance financial cover against impacts of natural





disasters (International Monetary Fund, 2019a). Insurance facilitates disaster risk resilience and adaptation by transferring residual risk away from individuals and communities. Cost effective and risk-informed sustainable development is based on the comprehensive understanding of hazards, the vulnerability of economies, societies and governments, and the exposure of society, people and belongings (UNDRR, 2019), but the lack of understanding of one or more of these components frequently limits the use of insurance mechanisms in many regions of the world most at risk from weather and climate hazards. This

leaves significant populations around the world more vulnerable to the economic consequences of events that are otherwise manageable in countries with well-developed insurance markets (von Peter et al., 2012).

Detailed understanding of hazards is an essential part of understanding risk, but simulation of tropical cyclones in the BoB remains challenging for the current generation of seasonal forecasting systems (Camp et al., 2015), global climate models

(Shaevitz et al., 2014) and reanalyses (Hodges et al., 2017), partly due the relatively course spatial and temporal resolution of the numerical simulations.  It is well understood that large-scale thermodynamics and vertical wind shear has a significant impact on TC intensity, but there are also numerous vortex, convective, turbulent and frictional dissipative processes (e.g. Bryan and Rotunno, 2009; Nolan et al., 2007; Tang et al., 2015 amongst others) that occur on much smaller scales and also influence TC intensity, the impacts of which are not captured in low resolution modelling. For example, extreme gusts

associated with vigorous (deep) convection will generally be under-estimated without kilometre scale grid spacing that can explicitly resolve deep convection (e.g. Leutwyler et al., 2017; Weisman et al., 1997). More generally, as summarised by Leutwyler et al. (2017, and references therein), grid spacings of O(1 km) are comparable to the size of the particularly energetic eddies in the planetary boundary layer. Consequentially, we expect that turbulent processes, as well as the dominant turbulent length scale, will still be under resolved in this 4.4km dataset.


Previous insights into TC hazards affecting Bangladesh focus on compiling catalogues of events (see Alam and Dominey-Howes, 2015 and references therein), or apply statistical analysis to event catalogues (e.g. Bandyopadhyay et al., 2018; Bhardwaj et al., 2020), and can only provide limited insight into the spatial extent, variability and magnitude of events based on first-hand eye-witness reports and limited observational records.  Other authors take a parametric wind-field approach,

combing the geostrophic (gradient) wind with a planetary boundary layer model to produce hazard maps at kilometre-scale resolution (e.g. Done et al., 2020; Krien et al., 2018; Tan and Fang, 2018); although this is a relatively computationally inexpensive approach, the quality of the result appears highly variable between global TC basins. Additionally, there are a number of holistic risk assessment views, that combe multiple sources of hazard data, recognising that there are multiple hazards associated with TCs, and that a combined risk assessment is non-trivial. However, these techniques are often limited

to particular events (e.g. Hoque et al., 2016, 2019) or particular areas (e.g. Alam et al., 2020).  In both cases, the quality of hazard and/or risk assessment is limited by available observational and track data.  In Bangladesh, a relatively sparse meteorological observational network and interrupted non-continuous data records place additional limits on the description of TC hazards.





In this study we make use of the latest generation Met Office regional model over the BoB at a grid-box resolution of 4.4km which has been used to generate ensemble simulations of 12 historical tropical cyclone cases (Steptoe et al., n.d.). The ensemble data provides spatially and temporally consistent, counterfactual simulations of these events (relative to well defined and observed TC cases), albeit limited by the constraints of the model configuration and computational resources. The ensemble configuration enhances our understanding of how each cyclone may evolve if a similar event were to happen again.

We combine the ensemble information in a spatially coherent manner to produce hazard maps at 4.4km resolution over Bangladesh for extreme wind (gust) hazards. Using Bayesian inference, we establish gust speed exceedance intervals (return periods) at 4.4km resolution across all of Bangladesh, and demonstrate how this information can be directly integrated into a decision making framework.

## 2 Geospatial processing

Tropical cyclone simulations are derived from a 9-member ensemble for 12 historical events, using the latest generation Met Office Unified Model (Brown et al., 2012) convection-permitting regional atmosphere configuration, based on Bush et al. (2020) – hereafter referred to as RA2. Further details of the regional modelling process and validation against the observations-based International Best Track Archive for Climate Stewardship (IBTrACS, Knapp et al., 2010, 2018) and ERA5 reanalysis (C3S, 2017) can be found in Steptoe et al. (n.d.). In general, median peak gust speeds from the regional model ensemble are

found to be 22 to 43 m s$^{-1}$ faster compared to ERA5 and IBTrACS. We use the ensemble output to first derive event 'footprints' – a common method within the catastrophe modelling community to define peak hazard relating to a given event. In this case, footprints are based on the maximum wind gust speed achieved within each model run of 48 hours, that implicitly collapses the time dimension to leave a 2D gust field in a longitude-latitude frame of reference. Although the original regional model data covers a significant portion of the BoB, we crop the data to approximately 87.5°E to 93.0°E and 20.5°N to 27.5°N.

### 2.1 Generalised Additive Modelling (GAM)

To condense information from all 9 regional model ensemble member footprints into a coherent spatial summary of the tropical cyclone hazard, we use a generalised additive model (GAM), after Hastie & Tibshirani (1986), based on the R package *mgcv* of Wood (2017), as a flexible spatial regression framework. GAMs are an extension of generalised linear modelling that use smooth functions of covariates to build a linear predictor and have previously been applied in similar geospatial natural hazard

assessments, such as storm count data over Europe (Youngman and Economou, 2017), spatial prediction of maximum wind speed over Switzerland (Etienne et al., 2010) and return level estimation for U.S. wind gusts (Youngman, 2019). In each case, these studies incorporate spatial information into the GAMs formation, thereby implicitly respecting the spatial interaction present in the source data, and use the spatial dependence as a source of information.



For our purposes, we use a Gaussian location-scale (GLS) model family (Wood et al., 2016) to describe the natural logarithm (log) of the gust speed, where both the mean and the log of the standard deviation are smooth functions of predictors – in this case, longitude and latitude. Although other model families were trialled (such as generalized extreme value and gamma distributions) the GLS family was found to have the best trade-off between computational efficiency and model fit. The general form of our GAM is:


$$y_i(s) \sim LogNormal(\mu(s), \sigma(s)^2)$$
$$f\mu(s) = f_\mu\big(lon(s), lat(s)\big)$$
$$log\big(\sigma(s)\big) = f_\sigma\big(lon(s), lat(s)\big)$$

where $y_i(s)$ is the response variable, namely log gust speed for each ensemble member $i$ in each grid cell $s = 1, \ldots, N$, $N = 207{,}081$. $f_\mu$ (a function of the mean) and $f_\sigma$ (a function of the variance) are each defined as thin-plate regression splines (Wood, 2003) – isotropic smooth functions of covariates $lon_i$ and $lat_i$ (longitude and latitude respectively). Each smooth function requires a user-defined maximum amount of desired flexibility (wiggliness), traditionally quantified by the number of "knots". This flexibility is objectively penalised within *mgcv* to avoid over-fitting, while optimally explaining the trends in the data (Wood, 2003). Trial and error shows that $\mathcal{O}(600)$ knots are required to construct thin-plate spline basis functions that

avoid over smoothing given the resolution of the regional model data. Under this model formulation, the mean $\mu(s)$ can be interpreted as an aggregated prediction across the ensemble members.

The smooth model parameters are estimated using restricted maximum likelihood (REML). However, once the model is fitted,

it can be shown that it has a Bayesian interpretation. In particular, the coefficients of the smooth functions are assumed to have a multivariate Normal prior distribution, whose covariance matrix determines the wiggliness penalisation (see Wood, 2017 for further details). A Gaussian approximation of the posterior distribution for the coefficients then provides a multivariate Normal distribution as the posterior (Gelman et al., 2013). In practice, once a GAM model is fitted to each named storm, under the Bayesian interpretation, we obtain 1000 simulations from the posterior distribution of the smooth function coefficients via

random draws from a multivariate normal distribution (MVN). The MVN mean vectors are the REML coefficient estimates, and the MVN covariance is derived as a function of the covariance matrix of the sampling distribution of the model coefficients. In Bayesian inference, sampling from the posterior distribution implies we can then derive samples from the posterior predictive distribution of gust speed for each grid cell, $y_i(s)$. The predictive distribution, a unique feature of Bayesian inference, fully quantifies estimation uncertainty and variability in gust speed across ensemble members. We take 1000

samples from the posterior predictive distribution and construct prediction intervals based on the empirical quantiles of these samples. To aggregate gust information from all ensembles of all named storms, we pool the 1000 posterior predictive





simulations from each event into a total of 12,000 samples from the predictive distribution of gust speed across all 12 events. Figure 1 summarises the key parts of this process.

Assessing the GAM specification for $y_i(s)$ with detrended quantile-quantile (worm) plots (based on the method of Augustin et al., 2012), Figure 2 shows that generally storms are well represented. For some storms (such as Aila, BOB01, BOB07, Bulbul, Rashmi & TC01B) there is a tendency for the GAM to overestimate the tails of the distribution (positive kurtosis) relative to the 4.4km data, as indicated by quantile-quantile plot points falling below the zero residual line. In these cases, the GAM will over-estimate extremes. Akash is the only storm where maximum gust speeds are likely to be underestimated in the

GAM relative to the 4.4km data, but only for extreme upper-tail gust speeds. Checking for the consistency of variance over the range of predictor values, shows that the distribution of the residuals is stationary for both longitude and latitude (not shown).

## 3 Tropical Cyclones in Bangladesh

Aggregating the 12 historical tropical cyclones ensembles, Figure 3 shows the 95th and 99th percentiles of the posterior

predictive maximum gust speed distribution across Bangladesh. Based on historical cases, the provinces of Chittagong, Barisal and Khulna are most exposed of high wind speed associated with tropical cyclone gusts, whilst Sylhet and Rajshahi are least exposed. The cities of Chittagong and Cox's Bazar are particularly at risk of maximum tropical cyclone gust speeds exceeding 45 m s$^{-1}$ (87 kn) and 60 m s$^{-1}$ (116 kn) respectively, in 5% of events making landfall. Maximum gust speeds in Dhaka are likely to reach 35 m s$^{-1}$ (68 kn) in 1% of events, and 25 m s$^{-1}$ (48 kn) in 5% of events. We note that despite the northern

provinces of Rajshahi, Rangpur and Mymensingh being over 200 km inland, they experience 95th and 99th percentile gust speeds greater than those observed in the populated provincial capitals of Dhaka, Barisal and Khulna. This reflects the influence of cyclones Fani (May 2019) and Aila (May 2009) which had strong persistent in-land tracks.

The gust speed hazard can also be considered in terms of the probability of exceeding a threshold. Using WMO thresholds for

tropical cyclone wind speeds (WMO, 2018), Figure 4 shows that significant areas of southern provinces (Khulna, Barisal and Chittagong) will experience maximum windspeed in excess of severe cyclonic storm condition ≥ 25 m s-1 (48 kn) with a probability of 20-50% per tropical cyclone event. At higher wind speeds, only areas within 30 km of the coastline are predicted to experience gust speeds in excess of very severe cyclonic storm conditions ≥ 33 m s-1 (64 kn) with the same likelihood (20-50% per event). Windspeeds in excess of super cyclonic conditions ≥ 62 m s-1 (120 kn) are predicted to be exceeded with a

likelihood of 0.5-5% per event in limited areas south of Chittagong, with a small area in the vicinity of Cox's Bazar seeing exceedances of 5-10% per event.



In addition to specific thresholds, exceedance probability curves (Figure 5) summarise information for gust speeds up to 80 m s$^{-1}$ (155 kn) for 18 of the most populated towns and cities in Bangladesh (grey lines) with four key cities highlighted. Coastal cities of Cox's Bazar and Chittagong are unsurprisingly the population centres most exposed to high gust speeds. Chittagong and Cox's Bazar are roughly 2.5 and 4.8 times more likely to experience tropical cyclones exceeding 'Very Severe' cyclonic storm conditions than Dhaka, for a landfalling cyclone.

### 3.1 Decision-making under uncertainty

By defining a loss function, it is possible to exploit the information in the Bayesian posterior predictive distributions to create a warning model based on decision theory (Lindley, 1991). Following Economou et al. (2016), defining a loss function $L(a,x)$ to quantify the consequences of the various actions $a$ (e.g. issuing warnings) that could be taken in the event of a landfalling TC of varying intensities $x$ (see Table 1 for an example of four discrete gust categories), provides a method of mapping predictive information onto an action. The optimum action $a*$, given some predictive information $y$ (i.e. predictions of gust speed $y_i(s)$ from the GAM), is one that minimises the loss $L(a, x)$ taking into account the uncertainty in the predictive information, expressed as the probability of TC intensity $x$ given predictive information $y$, $p(x / y)$:

$$a* = \arg\min \int_x L(a, x)\, p(x|y)\, dx$$

In practice, $p(x|y)$ can be easily computed from the predictive samples from the GAM, while the loss function $L(a, x)$ is defined subjectively. Defining $L(a,x)$ is a non-trivial process, as it should encapsulate the relative cost of false-positive (i.e. where action against a TC was taken, but the TC did not occur) and false-negative (i.e. where no action was taken, but the TC did occur) events. For the purposes of demonstrating the principle of this approach, we define a dummy loss function in Table 1, based on the four TC warning levels used in Bangladesh (WMO, 2018). Here relative loss is defined on a 100-point scale, where 0 equates to no loss associated with a given landfalling event, and 100 equates to maximum loss. Evacuation typically takes places at the 'Great Danger' level.

Figure 6 illustrates the optimal warning that should be issued based on Table 1 and the range of gust speed information summarised by our GAM. This can be interpreted as the default optimal action to take given an impending land-falling TC, and in reality it would be updated with every new forecast of the TC. Interpretation is strongly conditioned by the loss function, but it also results in a transparent workflow that clearly translates hazards into actions. In this case, the northern extent of TC risk, as highlighted in Figures 2 and 3, is again reflected in the warning level, but in practice separate loss functions could be defined for each province, or for different economic sectors of society. By understanding the exposure, vulnerability and decision-making process of each user of this information, bespoke warnings could be issued.



## 3.2 Limitations

Despite the ensemble simulation framework, our analysis is still restricted to only 12 historical cases, representing the recent 40-year period. The number of events was determined by the availability of source data for driving the regional model, for TC events that made landfall over Bangladesh – in this case limited to the period of ERA5 data availability, which at the time of analysis extended back to 1979. Given the relatively low ERA5 resolution, we selected events that achieving peak wind speeds of at least 33 m s$^{-1}$, to be sure they would be identifiable within the low-resolution ERA5 data.


The initial conditions posed in the regional model play a significant role in determining the outcome of each event. In forecasting situations this is desirable behaviour: well-chosen initial conditions ensure the model retains a realistic representation of reality. Even though the modelling domain that produced these 4.4km data had the freedom to deviate in a physically plausible way (see Steptoe et al., n.d.), it does not have the ability to sample the full spectrum of possible BoB

tropical cyclone events. Simulations driven by a wider range of initial conditions, derived from a wider range of historical cases, would improve the sample size of cyclonic conditions on which this analysis is based. Note that this wouldn't necessarily reduce uncertainty in exceedance thresholds (in a frequentist paradigm), but it would update our view (i.e. our posterior estimate) of what is credible within the continuum of possible tropical cyclone events. In Bayesian parlance, our posterior view of Bangladesh tropical cyclones would become our new prior belief if subsequent simulation data became

available.

A different limitation is posed by the initial aggregation of the 4.4 km model over time. This removes our ability to draw inferences on annual occurrence of (or longer-term variability in) TC events. This means that our estimates of exceedance probabilities are conditional on a tropical cyclone event actually impacting Bangladesh. For the purposes of risk assessment,

we do not feel this limitation is significant – current generation weather forecast models are capable of accurately predicting the landfall location and track of tropical cyclones in the BoB many days in advance. It should also be noted that due to the computation expense of the 4.4 km data simulation, only events that impact Bangladesh were assessed, so conclusions cannot be drawn on the frequency of other events within the wider BoB region.

## 4. Summary & Conclusions

Generalised additive models (GAMs) provide a useful framework for condensing spatial hazard information in an interpretable way, from multiple numerical model simulations, into a single spatially coherent hazard map. Using a restricted maximum likelihood approach to fit the GAM allows us to interpret model predictions in a Bayesian fashion that logically provides credible exceedance estimates. High-resolution convection-permitting numerical predictions of 12 historical cyclone events, in an ensemble model set-up, gives an improved sense of the plausibility and likelihood of possible extreme events given the
limited observational history in this region. Combining ensemble simulations with a GAM then allows us to robustly quantify

the likelihood of maximum gust speed exceedances in a spatially coherent manner.

Our new maps of exceedance intervals show that north-western provinces of Bangladesh are relatively exposed to high wind

speed hazards – in some areas the exceedance probabilities are equal to those experienced along the coast.  Our hazard-to-

decision making framework suggests that these areas may need to be considered in an equivalent manner to coastal regions,

from a disaster risk reduction perspective. In coastal areas of Cox's Bazar and Chittagong we show super cyclonic conditions

may occur as frequently as 1-in-20 to 1-in-100 years. We hope that these kilometre scale hazard maps facilitate one part of the

risk assessment chain to improve local ability to make effective risk management and risk transfer decisions. Future work to

co-produce a proper loss function, given wind speed thresholds, would facilitate a method of transparent operational decision

making that could be used as the basis of an operational warning system.

**Code availability**

Python, R and GAM model and data analysis code is available at https://doi.org/10.5281/zenodo.3953772

**Data availability**

The data used in this study is available at https://doi.org/10.5281/zenodo.3600201 and released under CC-BY 4.0

**Author contribution**

HS prepared the manuscript, with input from TE, and undertook the data analysis.  TE and HS jointly developed and coded

the GAM model. TE developed and coded the decision-making framework used in 3.1.

**Competing Interests**

The authors declare that they have no conflict of interest.

**Acknowledgements**

This study is part of the Oasis Platform for Climate and Catastrophe Risk Assessment – Asia (https://www.international-

climate-initiative.com/en/nc/details/project/oasis-platform-for-climate-and-catastrophe-risk-assessment-asia-18_II_165-

3018), a project funded by the International Climate Initiative (IKI), supported by the Federal Ministry for the Environment,

Nature Conservation and Nuclear Safety, based on a decision of the Germany Bundestag.




The authors thank Erasmo Buonomo, Richard Jones, Jane Strachan & Tamara Janes for comments that improved early versions of this manuscript.

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



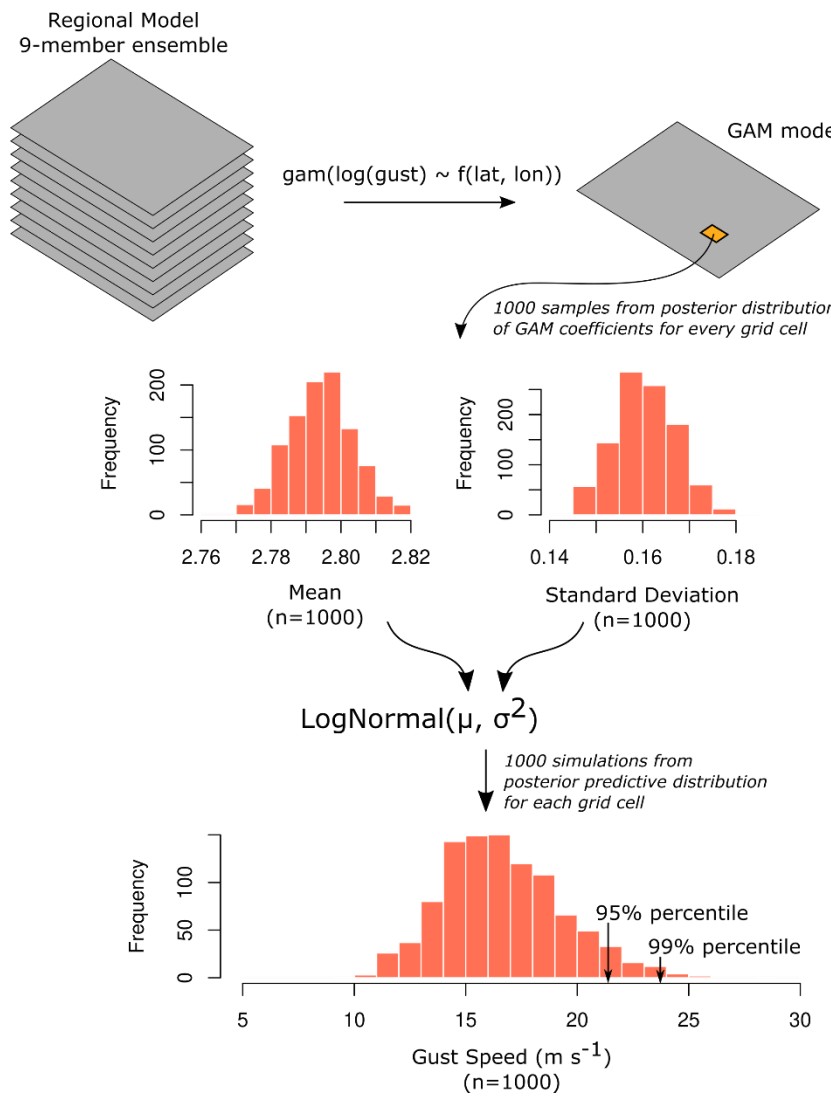


**Figure 1** Summary of generalised additive modelling and the derivation of the posterior prediction gust speed distribution. The posterior predictive distribution is found for each grid cell of the regional model domain. Gust speed prediction intervals are found from the percentiles of the posterior predictive distribution.





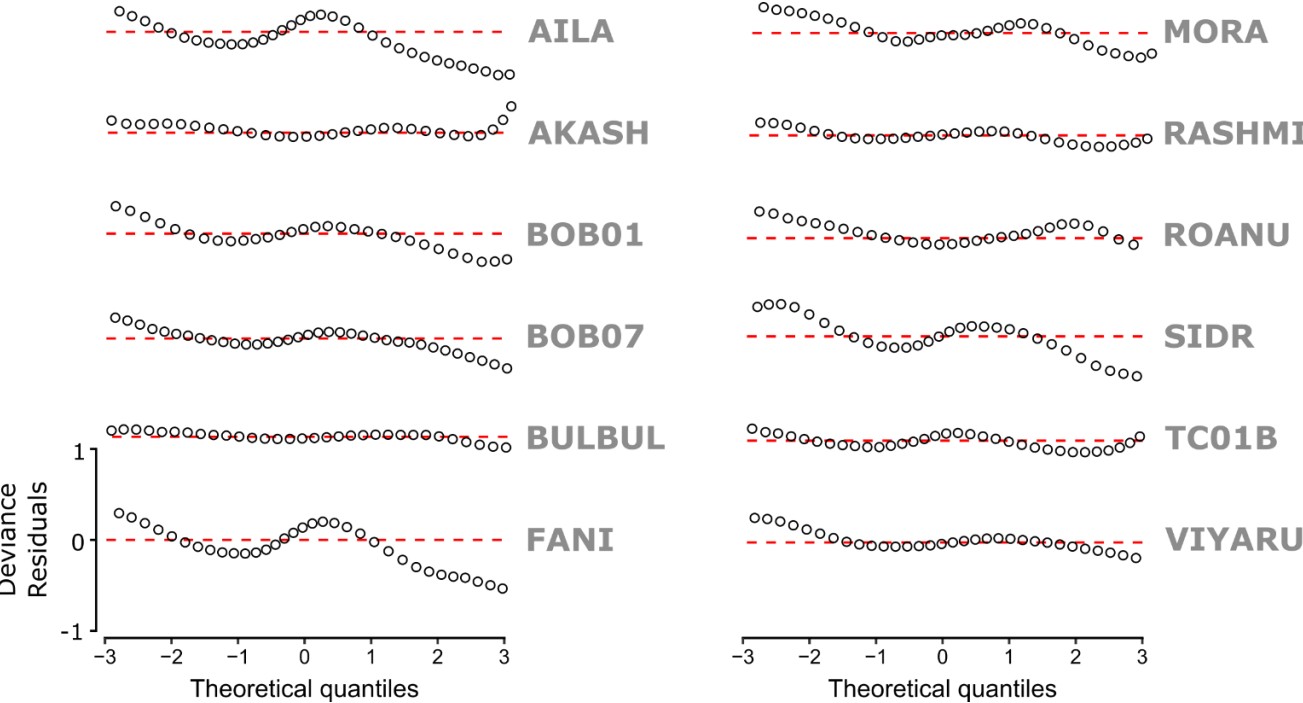

**Figure 2** Detrended quantile-quantile (worm) plots for each GAM model per storm. We discretise the quantiles into 50 bins (open circles). The red dashed lines represents zero deviance between data and theoretical quantiles defined in the GAM. Where model quantile deviates below (above) the zero deviance line, this implies that the model predictions are overestimated (underestimated) relative to the data: for any given theoretical model quantile, the data quantile is lower (higher). Deviance residuals respect the model family used when fitting the GAM and are calculated via the simulation method of (Augustin et al., 2012).



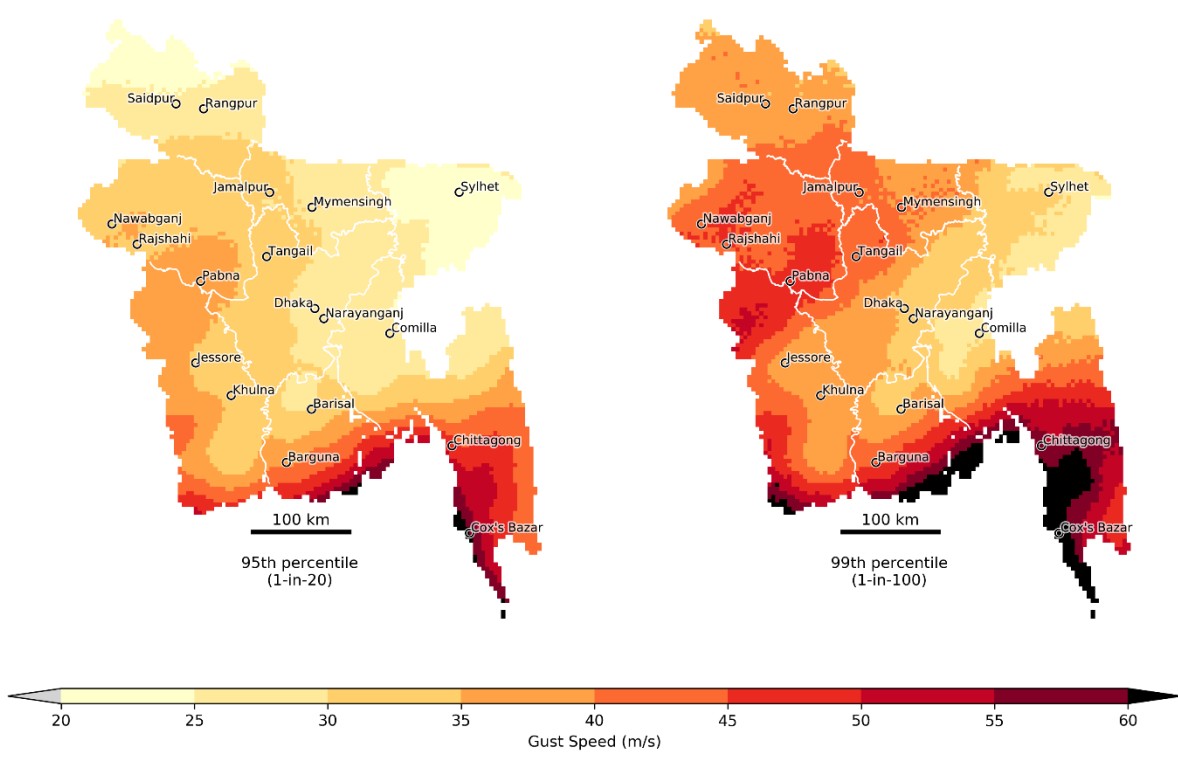

**Figure 3:** Gust speed exceedance thresholds for the 95[th] (left) and 99[th] (right) percentile credible intervals. The 95[th] and 99[th] percentiles represent the maximum gust speeds expected from a 1-in-20 and 1-in-100 event respectively (conditional on a tropical cyclone making landfall over Bangladesh). These credible intervals are based on the posterior model distribution derived from all 12 named tropical cyclones, conditional on a tropical cyclone making landfall in Bangladesh. The 20 – 60 m s-1 gust speed range roughly corresponds to a range of 39 – 117 kn, equivalent to the cyclonic to super cyclonic storm classification used in Bangladesh. Province boundaries are outlines in white.




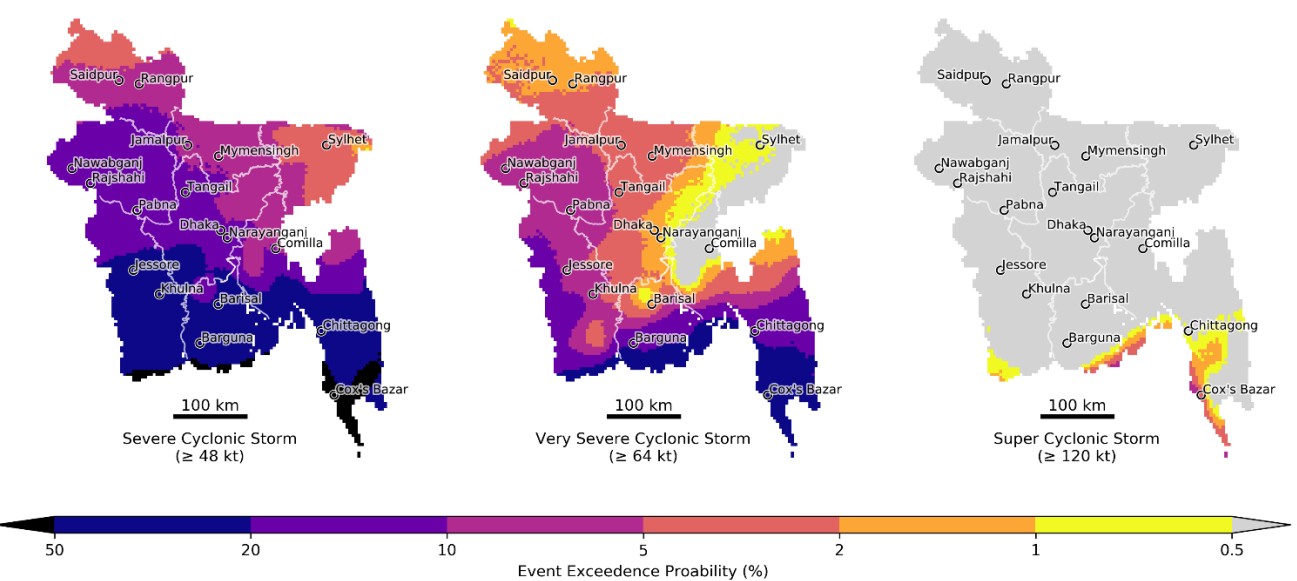

**Figure 4:** Event exceedance probabilities for a severe cyclonic storm (left), very severe cyclonic storm (middle) and super cyclonic storm (right) WMO tropical cyclone classifications used in the Bay of Bengal (WMO, 2018). Event exceedance probabilities show the likelihood of a maximum tropical cyclone gust speed being greater than or equal to the corresponding classification wind threshold, conditional on a tropical cyclone making landfall over Bangladesh. An exceedance threshold of 50% (0.5%) represent a 1-in-2 (1-in-200) chance of a tropical cyclone exceeding a given threshold. Areas where the exceedance probability is > 50% (< 0.5%) are shaded black (grey). Province boundaries are outlined in white.

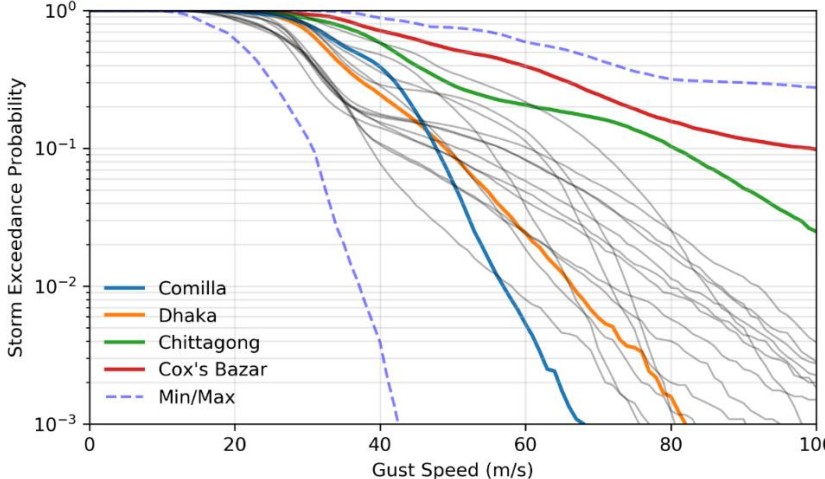

**Figure 5:** Exceedance probability curves for 18 of the most populated towns and cities in Bangladesh (grey lines), with 4 key cities highlighted: Dhaka (orange), Comilla (blue), Chittagong (green) and Cox's Bazar (red). For reference, the minimum and maximum range of exceedance probabilities (across all of Bangladesh) are represented by the dashed lines. Note that storm exceedance probability is shown on a log-scale.



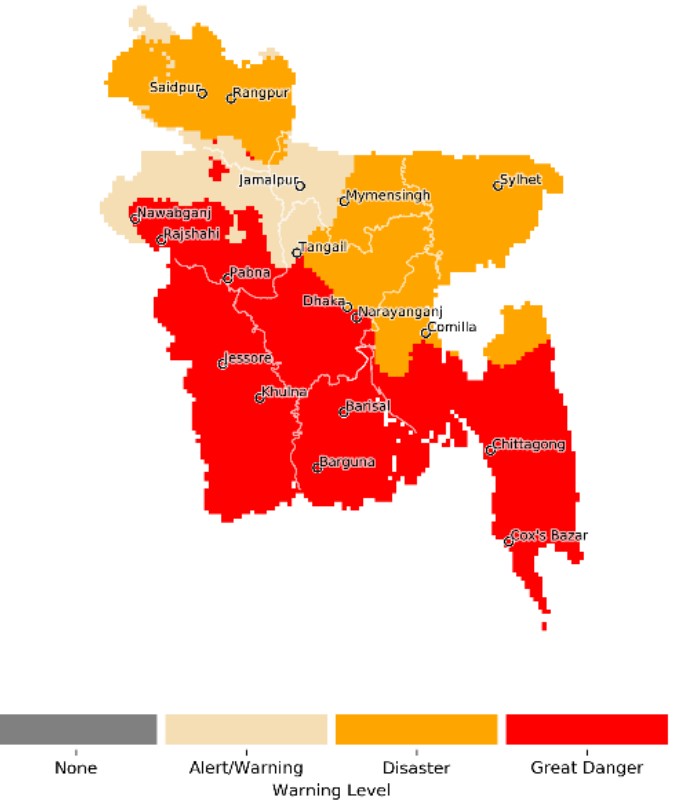

**Figure 6:** Example warning status given an impending landfalling tropical cyclone over Bangladesh. These warnings represent the most effective action minimising the loss as defined in Table 1.

| Loss Function | Warning Level (y) | | | |
|---|---|---|---|---|
| | OK | Warning | Disaster | Great Danger |
| < 50 km/h | 0 | 5 | 15 | 20 |
| 50 ≥ km/h < 61 | 50 | 10 | 20 | 25 |
| 61 ≥ km/h < 89 | 80 | 50 | 25 | 30 |
| ≥ 89 km/h | 100 | 100 | 80 | 40 |

390

**Table 1** Dummy loss function for actions associated with 4 Bangladesh TC warning levels, and their associated wind speed intensity. In this case loss is define on a 100-point scale, where 0 = no loss, and 100 = maximum loss, associated with a given landfall TC event.