# Peer review of "Extreme wind return periods from tropical cyclones in Bangladesh: insights from a high-resolution convection-permitting numerical model"

_Natural Hazards and Earth System Sciences, 2020_

## Referee Comment (RC1) · Anonymous Referee #1 · 5 Nov 2020

General comments

The authors present maps of exceedance intervals of extreme wind in Bangladesh, created using 9 ensemble members of 12 historic tropical cyclones. These maps are very clear and highlight the substantial risk facing north-western provinces, far away from the coast. Using return periods makes this study easy to interpret in the hazard risk community, where there is a lot of uncertainty around tropical cyclone risk. The authors use a novel tool of generalized additive models to condense 9 ensemble members into a coherent spatial summary. The authors pose an interesting scientific

question and show results that should be of interest to a wide and diversified audience.

This paper is very well written, and the figures are relevant and discussed appropriately in the results. The method comprehensive and clear, with mathematical formulae well defined. The technical aspects of the paper are handled well and should be understandable by fellow scientists. There is a detailed limitations section, which describes the problems with using model data and its availability.

This paper is a good length with no need for supplementary material. The references are accessible and sufficient in number, although Hersbach (2020) should be cited for ERA5.

The author gives proper credit to their contribution and that of the co-author (Line 235). The authors also include data analysis code and data availability statement.

The key conclusion is that "some northern provinces, up to 200 km inland, may experience conditions equal to or exceeding a very severe cyclonic storm event with a likelihood equal to coastal regions less than 50 km inland." The authors use sufficient data to come to these conclusions, with 12 historic storms and 9 ensemble members. Such a statement is important for disaster risk reduction.

Specific comments

Is it possible to remain consistent in the units? Figures 1, 3, and 5 use m/s, Figure 4 uses knots, and Table 1 uses km/h.

The title could be made more relevant, mentioning return periods / exceedance probability, and the generalized additive model, e.g. The use of generalised additive models to examine extreme wind return periods from tropical cyclones in Bangladesh.

Technical corrections

Space between: 4.4 km

Line 66: Add 9-member: "generate 9-member ensemble simulations of 12 historical

tropical cyclone"

Line 40: Change 'course' to 'coarse'

Line 58: Change 'combe' to 'comb'

Line 194: Include in brackets what the resolution of ERA5 is, as low for this study but not low in comparison to other reanalysis products. "Given the relatively low ERA5 resolution (x)"

Lines 210-211: Add a citation following this statement: "current generation weather forecast models are capable of accurately predicting the landfall location and track of tropical cyclones in the BoB many days in advance."

Line 141: Change 'of high wind speed' to 'to high wind speed'.

Note in the figures that the town labels are the 18 most populated.

Town names difficult to see in some of the figures.

Figure 2 caption: Change from "of (Augustin et al., 2012)" to "of Augustin et al. (2012)"

Figure 3: Superscript -1 for the units. Also change 'outlines' to 'outlined'.

Table 1: Change 'define' to 'defined'.

---

## Referee Comment (RC2) · Anonymous Referee #2 · 17 Nov 2020

This paper strikes me as technically interesting, but scientifically confused at worst, or badly explained at best. The authors have done ensembles of high-resolution simulations for 12 tropical cyclones, developed a statistical model to produce smooth maps of peak gust speed, and produced hazard maps that give the probabilities of given gust speeds conditional on a tropical cyclone making landfall in Bangladesh. The most thoroughly described aspect of the work is the statistical model. But the other essential aspects of the study are too deficiently explained for me to recommend publication. It might be possible to turn into a good paper, but the authors need to make clearer what

they are doing and why.

Major comments:

1. What is the study trying to achieve? Why is the problem formulated this way — what is the value of wind hazard maps conditional on a TC making landfall somewhere (anywhere) in Bangladesh?

In the introduction, the authors seem to be thinking about applications involving risk assessment, that is, knowing the long-term probabilities of a given hazard occurring. But for that application, one wants information that is not conditional on any short-term forecast; one wants the overall unconditional hazard. For this, 12 observed storms are not enough, I'd think, even with 9 ensemble members for each; one would want to span the whole space of possible storms. This normally involves looking at large numbers of synthetic storms, as in the catastrophe modeling done in the insurance industry.

Then, in the conclusions, the authors seem to be thinking about applications to short-term forecasting, rather different from the applications suggested earlier. But in that case, one would have an actual forecast of the specific storm that would be coming. Why would one want a general forecast only conditional on a TC anywhere in Bangladesh? Surely wind maps from a past storm whose center made landfall on the westernmost part of the coast, for example, are not relevant if the actual storm is heading to the easternmost part of the coast; but the maps produced here include both equally. It makes no sense to me.

I think the authors need to explain up front what problem they are trying to solve and why they are trying to solve it this way. I cannot actually see what useful problem justifies this particular approach. I might be missing something, but they need to make the argument more clearly.

2. The simulations are not described nearly enough. We know only the grid resolution. We aren't told where the domain boundaries are, the length of the integrations, etc. The

reader is referred to Steptoe et al. (n.d.) for this information. I don't know what "n.d." means, but this information is not available to the reader that I can see. At least the basics should be in the paper. Also we should see some comparison of the simulations to observations for these storms, so we have some idea how good the simulations are.

3. The statement that the median peak gust speeds simulated exceed those in IB-TRACS and ERA5 by 22-43 m/s is shocking — that is an ENORMOUS discrepancy. However the statement also makes no sense, or at least is not explained well enough. What is the definition of a gust in the model, in IBTRACS, and in ERA5, for example? To my knowledge actually neither IBTRACS nor ERA5 includes information on gusts, as normally defined, just "sustained", i.e., 1-min or 10-min average, wind speeds; a gust is usually defined as higher frequency, i.e., 3-second average or such. Is a gust in the model resolved or parameterized? Is it just the instantaneous wind speed at a time step on the model grid? If so, at what time resolution was it saved – every time step (and what is the model time step)? And once we have the answers to these questions, what level of agreement should one expect between the model and ERA5 or IBTRACS, given what I'm nearly certain are the very different natures of these data sets?

More minor, specific comments:

Line 75: How were these 12 storms chosen? Of what are they meant to be representative? What is the strategy here? See major comment 1 above.

Section 2.1: The authors go right into describing this somewhat sophisticated statistical model, but the reader at this point doesn't have enough of a clue what the objective is. "Condense information from all 9 regional model ensemble member footprints into a coherent spatial summary of the tropical cyclone hazard" is not enough. What is the reason to think these 12 storms x 9 ensemble members are representative enough for the purpose here, whatever that is? Can the authors please explain in plainer English what is being done here and why? And how can we determine if the answer we get is good or bad?

Line 147: isn't it a bit disturbing that the results are so strongly influenced by just two storms? Don't we want something more statistically robust than that?

Section 3.1: As in major comment 1, what is the problem being solved here? What can one use these results for that wouldn't be better served by a normal operational forecast simulation (or ensemble of same), which would have details of the specific storm?

Line 184: "and in reality it would be updated..." How so, how much would this change the answer, and again why do it this way?

Line 194: "to be sure they would be identifiable within the low-resolution ERA5 data." I don't understand this at all. Why do the storms need to be identified in ERA5? How is ERA5 data being used, other than to initialize the model? This is totally unclear.

---

## Author Comment (AC1) · 18 Nov 2020

We thank the reviewer for their helpful comments. Our replies are inline below:

**The references are accessible and sufficient in number, although Hersbach (2020) should be cited for ERA5.**
Corrected, thank you.

**Is it possible to remain consistent in the units? Figures 1, 3, and 5 use m/s, Figure 4 uses knots, and Table 1 uses km/h.**

Figure labels have been changed to use $m\ s^{-1}$ by default, with additional knot conversions in some cases. Table 1 has been converted to $m\ s^{-1}$.

**The title could be made more relevant, mentioning return periods / exceedance probability, and the generalized additive model, e.g. The use of generalised additive models to examine extreme wind return periods from tropical cyclones in Bangladesh.**
We will investigate the possibility of changing the title with the Editor.

**Space between: 4.4 km**
Corrected.

**Line 66: Add 9-member: "generate 9-member ensemble simulations of 12 historical C2 tropical cyclone"**
Updated.

**Line 40: Change 'course' to 'coarse'**
Corrected.

**Line 58: Change 'combe' to 'comb'**
This should be 'combine' – now corrected.

**Line 194: Include in brackets what the resolution of ERA5 is, as low for this study but not low in comparison to other reanalysis products. "Given the relatively low ERA5 resolution (x)"**
'(31 km)' added to this sentence.

**Lines 210-211: Add a citation following this statement: "current generation weather forecast models are capable of accurately predicting the landfall location and track of tropical cyclones in the BoB many days in advance."**
We have added citations of Mohanty et al., (2020) and Singh and Bhaskaran (2020) as examples.

**Line 141: Change 'of high wind speed' to 'to high wind speed'.**

[Figure]

Corrected.

**Note in the figures that the town labels are the 18 most populated. Town names difficult to see in some of the figures.**
'. . .with the 18 most populated towns and cities marked by circles.' added to the figure captions. We have made some subtle alterations to the label plotting whilst trying not to obscure to much of the underlying data.

**Figure 2 caption: Change from "of (Augustin et al., 2012)" to "of Augustin et al. (2012)"**
Corrected.

**Figure 3: Superscript -1 for the units. Also change 'outlines' to 'outlined'.**
Corrected.

**Table 1: Change 'define' to 'defined'.**
Corrected.

**Citations**
Mohanty, S., Nadimpalli, R., Mohanty, U. C., Mohapatra, M., Sharma, A., Das, A. K. and Sil, S.: Quasi-operational forecast guidance of extremely severe cyclonic storm Fani over the Bay of Bengal using high-resolution mesoscale models, Meteorol. Atmos. Phys., doi:10.1007/s00703-020-00751-4, 2020.

Singh, K. S. and Bhaskaran, P. K.: Prediction of landfalling Bay of Bengal cyclones during 2013 using the high resolution Weather Research and Forecasting model, Meteorol. Appl., 27(1), e1850, doi:10.1002/met.1850, 2020.

---

## Author Comment (AC2) · 20 Nov 2020

We thank the reviewer for their comments.

In general we note that the accompanying data validation paper (Steptoe et al., n.d.) is currently still under review with a different journal. As we feel this paper goes some way to addressing some of the concerns of the reviewer, we have attached it as a supplement to this reply.

Our replies are inline below:

[Figure]

**Major comments:**

**1. What is the study trying to achieve?**
Fundamentally, the paper aims to improve our understanding of wind (gust) extremes in Bangladesh from two perspectives: (1) improved spatial understanding, in terms of the spatial extent of wind extremes; (2) improved temporal understanding, in terms exceedance probabilities. Section 1 highlights a number of problems encountered in the current literature for this region, including: (1) a focus on the limited historical record (L51-54); (2) a focus on parametric wind field models that are computational expensive and often specifically tuned to particular TC basins (L54-57); and (3) take a holistic hazard approach, but which are then limited to examining specific events (L57-61). In all cases, the limited observational record adds additional constraints.

Section 3 summarises the state of gust hazards, based on 12 events, in terms of exceedance likelihoods and return periods and highlights the spatial variability of the hazard. We expect this information would be of use to long-term mitigation planning (on the order of years or decades). Section 3.1 focuses on a robust, transparent way to make decisions when information is uncertain.

**Why is the problem formulated this way - what is the value of wind hazard maps conditional on a TC making landfall somewhere (anywhere) in Bangladesh? In the introduction, the authors seem to be thinking about applications involving risk assessment, that is, knowing the long-term probabilities of a given hazard occurring. But for that application, one wants information that is not conditional on any short-term forecast; one wants the overall unconditional hazard. For this, 12 observed storms are not enough, I'd think, even with 9 ensemble members for each; one would want to span the whole space of possible storms. This normally involves looking at large numbers of synthetic storms, as in the catastrophe modeling done in the insurance industry.**
This paper addresses the hazard specific to one country. To make the most of our computationally expensive numerical modelling, it is necessary to constrain the hazard in question. In this case, rather than say looking at TCs in the Bay of Bengal, of which many TCs don't impact Bangladesh, we focus on those TCs that do specifically impact Bangladesh. This imposes the condition of our results being dependent on a TC impacting Bangladesh. Most, if not all, statistical models will have some degree of conditionality to their results, even if they are not explicitly stated – climate stationarity being a common implicit assumption, in the case of synthetic event sets from the catastrophe modelling community.

Quantifying the chance of extreme events is fundamentally constrained by the limited length of the observational record, and there is increasing understanding that events can occur in the current climate that have no observed precedence (e.g. Thompson et al., 2017, 2019). This underlies our interest in using a full numerical approach to simulating TCs. Our ensemble modelling approach provides the opportunity to explore plausible, yet unseen hazards. We agree that basing these simulations on 12 events is limiting, but the number of events was determined by the availability of source data (ERA5) – see our later reply for further discussion of this. Exploring the whole space of possible storms is the ultimate goal, but no methods currently achieve this.

**Then, in the conclusions, the authors seem to be thinking about applications to shortterm forecasting, rather different from the applications suggested earlier. But in that case, one would have an actual forecast of the specific storm that would be coming.**
The aim of the paper is to outline a method of improving the hazard-to-decision making process. The timeframes over which decision might be made could be hours, or day, or years. We are not clear which aspect of the conclusion (ie. Section 4) the reviewer feels specifically apply only to short term forecasting. The conclusion does specifically reference 1-in-20 and 1-in-100 year return periods, that may facilitate long-term disaster risk reduction planning.

The method of Section 3.1 is applicable across all time-scales, but we recognise that in this particular case (ie. relating to TC hazards) at very short time scales, weather

forecast information will probably supersede our generalised hazard assessment (as presented in Section 3). However, the decision making framework is still applicable even in a forecasting situation – it's just that the predictive hazard data (ie. p(x|y)) would be derived from a forecast rather than our hazard maps, which we note in L186-187.

**Why would one want a general forecast only conditional on a TC anywhere in Bangladesh?**
See our answer re. conditionality above.

**Surely wind maps from a past storm whose center made landfall on the western-most part of the coast, for example, are not relevant if the actual storm is heading to the easternmost part of the coast; but the maps produced here include both equally. It makes no sense to me. I think the authors need to explain up front what problem they are trying to solve and why they are trying to solve it this way. I cannot actually see what useful problem justifies this particular approach. I might be missing something, but they need to make the argument more clearly.**
We have edited the final paragraph of the Introduction to clarify these points:

"In this study we seek to improve our understanding of the latent extreme gust speed hazard associated with TCs. To address the lack of observation data in this region, we use the latest generation Met Office regional model over the BoB, to simulate 9 versions of 12 historical tropical cyclone cases representing 1979-2019. This generates spatially and temporally consistent, counterfactual simulations (relative to well defined and observed TC cases), albeit limited by the constraints of the model configuration and computational resources. This ensemble configuration enhances our understanding of how each cyclone may evolve if a similar event were to happen again. We combine the ensemble information in a spatially coherent manner to produce hazard maps at 4.4km resolution over Bangladesh for extreme wind (gust) hazards. Using Bayesian inference, we estimate gust speed exceedance intervals (return periods) across all of Bangladesh, and demonstrate how this information can be directly integrated into a

decision making framework."

**2. The simulations are not described nearly enough. We know only the grid resolution. We aren't told where the domain boundaries are, the length of the integrations, etc. The reader is referred to Steptoe et al. (n.d.) for this information. I don't know what "n.d." means, but this information is not available to the reader that I can see. At least the basics should be in the paper. Also we should see some comparison of the simulations to observations for these storms, so we have some idea how good the simulations are.**

Yes, we agree that validating the RA2 model output is important. Unfortunately, delays in the peer review process mean the accompanying paper Steptoe et al. (n.d.) is still under review, but perhaps we should have included it as a supplement to this submission. As noted above, we have now included it as a supplement to this reply. We have added some further information on the basic simulation set-up into Section 2 as follows:

"The RAL2 4.4km domain avoids placing model boundaries over the Himalayas and covers Nepal, Bhutan, Myanmar, most of India, and parts of the Tibetan plateau. To ensure model stability over this mountainous terrain, the RAL2 model was run with a 30 second time-step."

**3. The statement that the median peak gust speeds simulated exceed those in IBTRACS and ERA5 by 22-43 m/s is shocking - that is an ENORMOUS discrepancy. However the statement also makes no sense, or at least is not explained well enough. What is the definition of a gust in the model, in IBTRACS, and in ERA5, for example? To my knowledge actually neither IBTRACS nor ERA5 includes information on gusts, as normally defined, just "sustained", i.e., 1-min or 10-min average, wind speeds; a gust is usually defined as higher frequency, i.e., 3-second average or such. Is a gust in the model resolved or parameterized? Is it just the instantaneous wind speed at a time step on the model grid? If so, at what time resolution was it saved – every time step (and what is the model time**

**step)? And once we have the answers to these questions, what level of agreement should one expect between the model and ERA5 or IBTRACS, given what I'm nearly certain are the very different natures of these data sets?**

We feel the answers to these questions are all provided by the accompanying paper Steptoe et al. (submitted), which is currently going through the publication process with another journal. For transparency we have attached our submission as a supplement to this reply.

Yes, the differences between RA2 and ERA5 is large, but it is known that extreme gusts associated with vigorous convection in ERA5 are generally under-estimated, sometimes by a factor of two (Owens and Hewson, 2018). As you point out, IBTrACS doesn't explicitly provide gust speed information (although our accompanying papers also compares maximum wind speed), so we have amended the text to clarify this.

Addressing some of your specific concerns:

- Yes, IBTrACS only estimates wind speed. In Steptoe et al., we find IBTrACS maximum wind speed estimates are more similar to RA2 wind speed estimates, than ERA5 wind speed estimates (for details, please refer to the paper).

- ERA5 gusts are parametrised based on the 10m wind speed, friction velocity, atmospheric stability, roughness length and a convective contribution based on wind shear.

- RA2 model uses a gust parametrisation based on 10m wind speed with scaling proportional to the standard deviation of the horizontal wind that also accounts for friction velocity, atmospheric stability and roughness length.

Given the lack of reliable observations for this region, it is difficult to definitively say which dataset is 'best', however we feel there is enough agreement between the IB-

TrACS and RA2 wind speed (discussed in Steptoe et al.) that analysis of RA2 gust speeds will provide useful results for understanding gust extremes in this region.

**More minor, specific comments:**

**Line 75: How were these 12 storms chosen? Of what are they meant to be representative? What is the strategy here? See major comment 1 above.**
We highlight the limitation of 12 storms in Section 3.2. Specifically, the number of events was determined by the availability of source data (ERA5) for driving the regional model (RA2), for TC events that specifically impact Bangladesh. As ERA5 provides the initial conditions for our RA2 model, we are limited to the recent historical period from 1979-present (in this case, roughly August 2019, at which point we ceased any further RA2 simulations). We feel that these 12 events are representative of the recent historical period of TCs in Bangladesh.

**Section 2.1: The authors go right into describing this somewhat sophisticated statistical model, but the reader at this point doesn't have enough of a clue what the objective is. "Condense information from all 9 regional model ensemble member footprints into a coherent spatial summary of the tropical cyclone hazard" is not enough. What is the reason to think these 12 storms x 9 ensemble members are representative enough for the purpose here, whatever that is? Can the authors please explain in plainer English what is being done here and why? And how can we determine if the answer we get is good or bad?**
We agree that ideally we would have access to more that 9 versions of 12 events, but observational records are poor for Bangladesh. IBTrACS data only provides information along a trajectory, and ERA5 data does not have sufficient resolution to capture TCs properly. We feel that our 108 simulations are the best possible source of data at the current time. Some elements of the 'good or bad' assessment of these simulations is done in Steptoe et al. (n.d.), which is now attached as a supplement for your information. We have added additional clarification around the purpose in the Introduction, as per our comments in reply to your Major comment 1.

**Line 147: isn't it a bit disturbing that the results are so strongly influenced by just two storms? Don't we want something more statistically robust than that?**
There are very few historical events that track over the northern part of Bangladesh, so we are not surprised at this result. Given the available information, this is the best estimate we can make of the potential for high gust speeds in these regions, but note that the posterior predictive distribution for grid cells in these locations still contains information derived from the other 10 events. Looking at the 95th and 99th percentile gust speeds will inevitably reflect the extreme events in any given location, but these events will be rare occurrences.

We allude to the statistical robustness (or otherwise) of the predictive distribution in L203-L208. Specifically note that under Bayesian inference, sampling a wider range of historical cases would reallocate the credibility of extreme gust speeds across possibilities.

**Section 3.1: As in major comment 1, what is the problem being solved here? What can one use these results for that wouldn't be better served by a normal operational forecast simulation (or ensemble of same), which would have details of the specific storm?**
The aim of this section is to demonstrate a method for making decisions under uncertainty. A 'normal operational forecast' simulation has uncertainty associated with it (irrespective of whether this is communicated to the user of the forecast information). The problem is: 'how to make the optimal decision given the uncertainty'. This is a concept addressed by Decision Theory that this Section explores in the context of our previously defined predictions of gust speeds, derived from our statistical (GAM) model. An operational forecast (or ensemble of forecasts) does play an important role, which is the point we were trying to make in L184 (as per your next comment below).

In the context of exposure to a hazard and being able to plan and mitigate against vulnerability, we think that it is an advantage to understand the optimal action given the historical hazard information. This is what Figure 6 shows. With new information (such

as a specific storm forecast) this optimal action would (probably) be superseded, but the method of translating the (new) hazard information into action is exactly the same.

**Line 184: "and in reality it would be updated..." How so, how much would this change the answer, and again why do it this way?**

The warning level is a function of the posterior predictive distribution obtained from the GAM, and the loss function defined in Table 1. The extent to which new information (eg. a new event forecast) would change the 'answer' depends on how much the new forecast deviates from the posterior predictive distribution and the confidence you give this forecast. We have added words to clarify these points, but note that aim of this section is to demonstrate a transparent workflow that clearly translates hazard information into actions.

**Line 194: "to be sure they would be identifiable within the low-resolution ERA5 data." I don't understand this at all. Why do the storms need to be identified in ERA5? How is ERA5 data being used, other than to initialize the model? This is totally unclear.**

As ERA5 data is used to initialise the RA2 model, TC features must be present in order for the RA2 simulation to proceed. It is possible that for small scale, or weak cyclogenic features, the resolution of the ERA5 dataset will smear these out. Trying to feed these sorts of events into the RA2 model would result in nothing being simulated. Hence, we focus on known events that have peak wind speeds of at least 33 m s-1 (ie. a Category 1 TC as defined on the IBTrACS website). We have added some words to this section to clarify these points.

**Citations**

Owens, R. G. and Hewson, T. D.: ECMWF Forecast User Guide, ECMWF, Reading, UK., 2018.

Steptoe, H., Savage, N., Sadri, S., Salmon, K., Maalick, Z. and Webster, S.: Tropical cyclone simulations over Bangladesh at convection permiting 4.4km 1.5km resolution,

[Figure]

Sci. Data, (submitted), n.d.

Thompson, V., Dunstone, N. J., Scaife, A. A., Smith, D. M., Slingo, J. M., Brown, S. and Belcher, S. E.: High risk of unprecedented UK rainfall in the current climate, Nat. Commun., 8(1), 1–6, doi:10.1038/s41467-017-00275-3, 2017.

Thompson, V., Dunstone, N. J., Scaife, A. A., Smith, D. M., Hardiman, S. C., Ren, H.-L., Lu, B. and Belcher, S. E.: Risk and dynamics of unprecedented hot months in South East China, Clim. Dyn., 52(5–6), 2585–2596, doi:10.1007/s00382-018-4281-5, 2019.

Please also note the supplement to this comment:
https://nhess.copernicus.org/preprints/nhess-2020-299/nhess-2020-299-AC2-supplement.pdf

**Supplement:**

**Tropical cyclone simulations over Bangladesh at convection permitting 4.4km & 1.5km resolution**

Hamish Steptoe[1], Nick Savage[1], Saeed Sadri[1], Kate Salmon[1], Zubair Maalick[1], Stuart Webster[1]

5    [1]Met Office, FitzRoy Road, Exeter, EX1 3PB, UK

*Correspondence to*: Hamish Steptoe (hamish.steptoe@metoffice.gov.uk)

**Abstract.** High resolution simulations at 4.4km and 1.5km resolution have been performed for 12 historical tropical cyclones impacting Bangladesh. We use the European Centre for Medium-Range Weather Forecasting 5[th] generation Re-Analysis (ERA5) to provide a 9-member ensemble of initial and boundary conditions for the regional configuration of the Met Office

10    Unified Model. The simulations are compared to the original ERA5 data and the International Best Track Archive for Climate Stewardship (IBTrACS) tropical cyclone database for wind speed, gust speed and mean sea-level pressure. The 4.4km simulations show a typical increase in peak gust speed of 41 to 118 knots relative to ERA5, and a deepening of minimum mean sea-level pressure of up to -27 hPa, relative to ERA5 and IBTrACS data. Generally, the timing of gust maxima and mean sea-level pressure (MSLP) minima are delayed relative to ERA5 and IBTrACS. Cyclones in the 1.5km dataset have similar MSLP

15    minima, but slightly faster maximum gust speeds. The downscaled simulations compare more favourably with IBTrACS data than the ERA5 data suggesting tropical cyclone hazards in the ERA5 deterministic output may be underestimated. The dataset (Steptoe et al., 2020) is freely available from https://doi.org/10.5281/zenodo.3600201.

**1 Introduction**

To construct this dynamically simulated tropical cyclone dataset dataset we use the latest generation Met Office regional

20    models to simulate tropical cyclones (TCs) over the Bay of Bengal (BoB) at grid-box resolutions of 4.4km and 1.5km. Using the ERA5 reanalysis data (C3S, 2017; Hersbach et al., 2018) to initialise and provide boundary conditions for our regional models, we dynamically downscale 12 historical TCs that made land-fall over Bangladesh between 1991 and 2019, using an ensemble approach.

25    Downscaling of ERA5 is reported in a few other studies: Bonanno et al. (2019) downscale ERA5 using the Weather Research and Forecasting (WRF) model to produce a new 7km reanalysis over Italy; preliminary work by Taddei et al. (2019) use ERA5 to force the BOlogna Limited Area Model-MOdello LOCale (BOLAM-MOLOCH) regional model for the purposes of coastal risk assessment in the North Western Mediterranean sea, and Wang et al. (2020) use ERA5 to run a 10km WRF domain over high mountain Asia. Specifically examining tropical cyclones, many studies use variations of the Weather Research and

30 Forecasting (WRF) Model (e.g. Skamarock et al., 2019), such as Kaur et al. (2020) who use WRF to downscale the National Center for Environment Prediction (NCEP) Climate Forecast System (CFSv2) and its atmospheric component Global Forecast System (GFS) to 9km over the north Indian Ocean for two historical cases (Mora and Ockhi), with analysis focusing on the spatial accuracy of rainfall and 850 hPa vorticity, and the vertical profiles of wind and temperature. They conclude that the downscaled model significantly improves the spatial distribution of rainfall, maximum vorticity evolution, wind, and temper-

35 ature profiles for mature phase cyclones. Studies specifically examining the BoB simulations include Srinivas et al. (2013), Singh & Bhaskaran (2020) and Mahala et al. (2019), amongst others. These studies typically make empirical comparisons of TC simulations at ~10km resolution against observationally based data, but often with an India-centric domain that contains a larger number of landfalling events. By contrast, in this study we specifically focus on Bangladesh, with simulations at higher resolution.

40 We make 12 variables available, including: air temperature, maximum wind gust speed, minimum air pressure at sea level and precipitation amounts (see Table 1), at a range of temporal scales (including model instantaneous values) as well as hourly and daily aggregations. Simulations are performed for the following tropical cyclones (landfall date): BOB01 (Apr 1991), BOB07 (Nov 1995), TC01B (May 1997), Akash (May 2007), Sidr (Nov 2007), Rashmi (Oct 2008), Aila (May 2009), Viyaru (May 2013), Roanu (May 2016), Mora (May 2017), Fani (May 2019) and Bulbul (Nov 2019). Table 2 lists approximate landfall

45 times and their International Best Track Archive for Climate Stewardship (IBTrACS, Knapp et al., 2010, 2018) ID number. At the time of writing, ERA5 data are only available from 1979 onwards, so our new catalogue excludes cyclones prior to 1979, most notably Cyclone Bhola of November 1970. Section 2 describes the RAL2 numerical model, the storm tracking algorithm and the key aspects of ERA5 and IBTrACS datasets, and we compare our results to the source ERA5 reanalysis and the International Best Track Archive for Climate Stewardship (IBTrACS) tropical cyclone database v.4 (Knapp et al., 2010,

50 2018) in Section 3.

**2 Methods**

**2.1 Numerical modelling**

Our high-resolution convection-permitting modelling utilises the latest generation Met Office Unified Model (Brown et al., 2012) v11.1, regional atmosphere configuration RAL2-T, a further development of RAL1-T (after Bush et al., 2020) – hereafter

55 referred to as RAL2. For each historical tropical cyclone case listed on Table 2, we run the RAL2 model in a 'downscaling' configuration, using ERA5 data to initialise and provide boundary conditions for a series of 9 time-lagged ensembles (see Figure 1 for a visual representation of this configuration).

As there is no data assimilation process or nudging, the initial conditions imposed by ERA5 are found to have significant

60 influence on the resulting tropical cyclone development. The time-lagged configuration is designed to limit the free-running

model time to 72 hours, whilst ensuring that the central 24-hour period of interest (centred on the tropical cyclone landfall time) is sufficiently sampled from a range of ERA5 initial conditions. This initial condition ensemble approach produces a set of 9 plausible tropical cyclone development scenarios associated with each named event. After initialisation, each ensemble member is free running for 72 hours, with hourly boundary conditions provided by ERA5. Each run requires a 24-hour spin-up period as the regional model adjusts from the weak initial state inherited from the ERA5 driving global model. This initial 24 hours of model data are discarded in subsequent analysis and data files. Together, the amassed ensemble provides 9 simulations of the central 24 hours, but covers a total period of 72 hours.

The RAL2 4.4km domain avoids placing model boundaries over the Himalayas and covers Nepal, Bhutan, Myanmar, most of India, and parts of the Tibetan plateau; the RAL2 1.5km domain is limited to Bangladesh only (Figure 2). To ensure model stability over this mountainous terrain, the RAL2 model was run with a 30 second time-step for both 4.4km and 1.5km simulations with additional orographic smoothing applied (using a 1-2-1 filter) to model cells 1500m above mean sea level.

**2.2 Storm tracking**

Storm tracking is performed on 3-hourly fields of RAL2 mean sea-level pressure (MSLP), 400 hPa temperature and 10m wind speed, using the Tempest extremes software of Ullrich and Zarzycki (2017). Unfortunately, 3-hourly fields are not frequent enough to estimate landfall time using the Tempest tracking algorithm for RAL2 data. The tracking algorithm has two parts – the initial feature detection and the stitching of these features to calculate tracks.

Feature detection is based on finding minima in air pressure at sea level, with features within a radius of 6° of each other being merged. The features are then further refined with a two 'closed contour criteria'. First an increase in sea level pressure of at least 200 Pa (2 hPa) within 5.5° of the candidate node, and second a decrease in 400 hPa air temperature of 0.4 K within 8° of the node within 1.1° of the candidate with maximum air temperature.

Stitching, to combine the individual features into tracks, uses a maximum distance between features of 3°, a minimum track length of 2 points (equivalent to 6 hours) and a minimum path distance of 0.1°. We also apply a topographic filter and a filter on maximum wind speed: tracks were rejected if they did not have at least one time-step and last at least 24 hours at an altitude less than 10m; and if they did not have maximum wind speed of at least 17 m s$^{-1}$ at one time-step.

**3 Datasets**

**3.1 ERA5 Reanalysis Data**

90   ERA5 (C3S, 2017; Hersbach et al., 2018) is the fifth and latest generation reanalysis dataset issued by the European Centre for Medium-Range Weather Forecasts (ECMWF). It combines both model data and observations on a real-time basis in a data assimilation process. Like a forecast, newly available observations are combined with model data to produce the best estimate of the state of the atmosphere. ERA5 data offers many improvements on the previous reanalysis, ERA-Interim, including more developed model physics and dynamics and an increased horizontal resolution of 30km. In term of vertical resolution

95   and extent, it has 137 model levels up to 80km.

For ERA5, we compare our simulated storm data with '10 metre wind gust since previous post-processing' defined as the maximum 3-second wind for each hour (parameter ID 49) and MSLP (parameter ID 151). Prior to 30th Sep 2008, ERA5 gust estimates only include turbulent contributions; the convective contribution was added to the wind gusts in post-processing for

100   events after this date (Bechtold and Bidlot, 2009).

**3.2 International Best Track Archive for Climate Stewardship (IBTrACS)**

International Best Track Archive for Climate Stewardship (IBTrACS, Knapp et al., 2010, 2018) forecasts are made by numerous forecasting centres around the world, and consists of the positions and intensities of tropical cyclones (Kruk et al. 2010). For our validation purposes, two Regional Specialized Meteorological Center (RSMC) datasets are used: the India Meteoro-

105   logical Department, New Delhi (IMD), and the Central Pacific Hurricane Center, Honolulu (CPHC).

IBTrACS best track data are typically calculated using a post-season reanalysis of storm positions and intensities from all available data, including ship, surface and satellite observations (Kruk et al. 2010). Typically, best track data consist of a time series of the storm's position, maximum sustained wind speed (in knots) and minimum central pressure. Estimated uncertainty

110   of the IBTrACS forecast wind speed are ±10 to ±20 knots, with positional uncertainty radiuses of 10km to 40km, dependent on wind speed intensity (IBTrACS, 2019). No uncertainty information is provided for pressure, but we note that the World Meteorological Organisation typically assume reporting precision of ±3 hPa. We also note that IBTrACS data is subject to forecaster best judgement and best track data typically lags the provisional operational data cyclone estimates by some months, subject to the availability of reanalysis data.

115

For the IBTrACS dataset we compare with 'maximum sustained wind speed' and MSLP. Although the WMO(1983) defines sustained wind speed as a 10-minute average windspeed at 10-m height above ground, it is reported as 1-minute averages by US forecast centres, and 3-minute averages by IMD. Some agencies, including CPHC, estimate gust speeds; however this data is not available for the BoB basin. Methods for obtaining maximum wind speed in IBTrACS vary by agency, as do their

120 availability of TC observation data. IBTrACS minimum central pressure is generally estimated with both subjective and objective satellite analysis as well as automated buoys that may be present (IBTrACS, 2019). Note that IBTrACS estimates usually end once the cyclone makes landfall.

**3.3 Comparing Datasets**

For the purposes of comparing RAL2 simulated winds and gusts with IBTrACS and ERA5, the RAL2 maximum sustained
125 wind speed is taken as the maximum of a single RAL2 model timestep windspeed over the accumulation period (1 hour). This is broadly comparable to a sustained maximum windspeed calculated with 30-second averaging period. In contrast, the parameterised RAL2 gust diagnostic represents a prediction of the 3-second average windspeed at every timestep. The maximum of this 3-second average speed over an hour is then taken to give the hourly maximum 3-second gust speed.

130 Considering the ERA5 and RAL2 model physics, ERA5 uses a mass flux scheme for cumulus parameterisation (an updated version of Tiedtke, 1989) whereas RAL2, while not truly resolving deep convection, is able to explicitly represent deep convective processes within the resolved dynamics. At these kilometre-scale resolutions the lower horizontal size limit of convective cells is still set by the effective resolution (e.g. 1.5km or 4.4km). More generally, as summarised by Leutwyler et al. (2017, and references therin), only grid spacings on the order of 1km are comparable to the size of particularly energetic eddies
135 in the planetary boundary layer, so the turbulent processes as well as the dominant turbulent length scale will be under resolved in both our downscaled model and ERA5. ERA5 gusts are parametrised based on the 10m wind speed, friction velocity, atmospheric stability, roughness length and a convective contribution based on wind shear between the model levels at 850hPa and 925hPa (Bechtold and Bidlot, 2009). It is known that extreme gusts associated with vigorous convection in ERA5 are generally under-estimated, sometimes by a factor of two (Owens and Hewson, 2018). The RAL2 model uses a gust parametri-
140 sation based on 10m wind speed with scaling proportional to the standard deviation of the horizontal wind that also accounts for friction velocity, atmospheric stability and roughness length (see Lock et al., 2019 for further details).

Comparisons of minimum MSLP are more straightforward. We compare the RAL2 hourly minimum MSLP estimated every 30-seconds, with the hourly minimum MSLP from ERA5, and the 3-hourly minimum MSLP from IBTrACS.

145 **4 Data Validation**

A lack of reliable, high-frequency and consistent meteorological observation data available for Bangladesh mean that verification of modelling results against in-situ observational data is not possible. Instead we establish the validity of the RAL2 4.4km data relative to ERA5 and the IBTrACS catalogue. It is important to recognise the differences in how the data are collected, their processing and resolution (see Table 3). Comparison of storm tracks is performed against the IBTrACS best
150 track data, after Kruk et al. (2010), only.

For the purposes of validation, we focus on three key variables: maximum wind speed, maximum gust speed and minimum pressure at mean sea level (MSLP). All comparisons against IBTrACS compare hourly maximum wind from our RAL2 4.4km model versus 3-hourly maximum wind speed estimates from IBTrACS. For maximum gust speed, we compare the RAL2

155    hourly maximum 3-second gust diagnostic with ERA5 hourly maximum 3-second gust speed diagnostic. MSLP estimates are comparable across all three datasets. In each case, the comparison is performed over a land-masked longitude-latitude domain that extends [79, 100]°E and [10, 25]°N – see Figure 2. This domain explicitly seeks to focus on the Bay of Bengal so as to compare model fields without land effects. In all cases, excluding land areas has very minor impact on the validation comparison (not shown) as peak wind, gust and minimum MSLP all occur over the ocean. Although our storm tracking output does

160    not allow us to explicitly compare the time of landfall between datasets (see Section 2.2), we expect that differences in the time of peak wind speeds would be mirrored in the differences in the time of landfall across datasets as peak wind speeds tend to occur just prior to landfall.

Each validation plot (Figure 3, and Appendix B) displays the gust speed, wind speed and MSLP from the ERA5, IBTrACS

165    and RAL2 4.4km. We resample the IBTrACS 3-hourly data by forward filling to 1-hourly intervals to aid the comparison of max/min timing with ERA5 and RAL2 datasets. Where IBTrACS maxima (minima) persist over several hours, the time differences reported in Sections 4.1 and 4.2 are then the minimum time difference between padded IBTrACS data and RAL2. The actual difference of RAL2 with respect to ERA5 (RAL2 – ERA5) is denoted ΔERA5 for brevity. For IBTrACS, actual differences with respect to IMD and CPHC are denoted ΔND and ΔUS respectively.

170

The statistical robustness of differences between datasets are assessed using the percentile bootstrap hierarchical shift function (Rousselet, G. A. and Wilcox, 2019; Rousselet et al., 2017) based on Wilcox & Erceg-Hurn (2012) and Wilcox et al. (2014). Given the potential skewness of the data, rather than looking at the differences of a single estimate of central tendency across all events (e.g. the median), differences are assessed for deciles (or percentiles) across the full distribution of the data, calcu-

175    lated using the distribution-free Harrell-Davis estimator (Harrell and Davis, 1982). This method explicitly deals with the hierarchical setting of data representing the same event, sharing common synoptic atmospheric conditions, but where different events are independent in time. The robustness of differences is assessed using bootstrapped (n=1000) uncertainty intervals for each decile difference. Where the 95% highest density interval (HDI) of uncertainty does not intersect zero, decile differences are considered statistically robust.

180    **4.1 Intensity and timing of maximum sustained wind speed**

For all events, RAL2 maximum sustained wind speeds are faster than ERA5 wind speeds (Figure 4), with median (across all events) ΔERA5 = 35 kn (18 m s$^{-1}$), with the 5$^{th}$ to 95$^{th}$ percentiles of the data spanning [10, 70] kn ([5, 36] m s$^{-1}$). Comparing IBTrACS, median ΔUS = -6 kn (-3 m s$^{-1}$) and ΔND = 10 kn, (5 m s$^{-1}$). Assessing the robustness of differences, the distribution

of ΔERA5 is robustly slower than RAL2 across all deciles (based on 95% HDI for each decile difference). ΔND is also robustly
slower for differences greater than the 40[th] percentile; however, note that at the time of writing, IBTrACS IMD maximum
sustained wind speed data for Fani and Bulbul were unavailable. Although IBTrACS US data has a tendency toward faster
sustained wind than RAL2 (i.e. negative ΔUS) these differences are not robustly different to zero at the 95% HDI.

The timing of maximum wind speed shows significant variation between events, with no clear correlation to peak wind inten-
sity differences; however, generally RAL2 peaks are delayed relative to ERA5 and IBTrACS data. Across all events, median
ΔERA5 = 5.5 hours delay, with ΔUS = 2.5 hours and ΔND = 0.5 hours. Only ΔERA5 and ΔUS times are robustly different to
RAL2 (evaluated at the 95% HDI). The largest time differences occur against ERA5 data: e.g. for Fani, some RAL2 ensemble
members show maximum wind intensities delayed by over 20 hours relative to ERA5 (see also Figure A5), but it is noted that
for these cases the ERA5 tropical cyclone simulation seems especially weak (for maximum wind, gust and minimum MSLP)
compared to IBTrACS data. Some of the variance in peak times will also derive from the differences in data frequency (1-
hourly for RAL2 versus 3-hourly for IBTrACS) but this requires further investigation to quantify.

**4.2 Intensity and timing of mean sea-level pressure**

For most events, the RAL2 ensemble produces deeper MSLP minima than the ERA5 and IBTrACS data (Figure 5), but whilst
ΔERA5 (median = -18 hPa) and ΔND (median = -10 hPa) differences with RAL2 are robustly different to zero, ΔUS (median
= -2 hPa) is not (all evaluated at the 95% HDI). At the time of writing, IBTrACS MSLP data for Fani and Bulbul are unavail-
able from IMD, and BOB01, BOB07 and TC01B are unavailable from CPHC.

As for wind speeds, the timing of RAL2 MSLP minimum is typically delayed relative to IBTrACS or ERA5 data. Median
time difference of MSLP minima are similar to wind speed maxima differences: ΔERA5 = 7.5 hours delay, ΔUS = 3.5 hours
and ΔND = 0.5 hours. Again, only ΔERA5 and ΔUS times are robustly different to RAL2 (evaluated at the 95% HDI). As for
the timing of gust peaks (Section 3.1), the RAL2 simulation of Fani shows median delays in MSLP minima of 14 hours
(ΔERA5) and 11 hours (ΔUS). BOB01 also has an equivalent delay of 13 to 14 hours (ΔND and ΔERA5 respectively).

**4.3 Intensity and timing of maximum 3-second gust speed**

The distribution of RAL2 gust speeds across events, are uniformly higher than ERA5 (Figure 6). The median difference across
all events is 63 kn (32 m s[-1], Figure 6), with some particularly strong individual events showing median differences up to 93
kn (48 m s[-1], BOB01) and 118 kn (61 m s[-1], Sidr). Comparing differences in the RAL2 and ERA5 gust speed distributions
using bootstrapped median difference by percentile across all events, shows that these differences are robustly different to zero
at the 95% HDI.

215 As with wind and MSLP, differences in the timing of maximum 3-second gust speed vary considerably between events with no clear correlation between the magnitude of the gust difference and the absolute time differences. The median time difference across all events is 2.5 hours (Figure 6), but this is not robustly different to zero at the 95% HDI.

**4.4 Storm Tracks**

We compare the track density of our nine downscaled ensemble members to IBTrACS in 30x30km spatial bins. Typically,
220 the area influenced by the tropical cyclone wind hazard is in excess of 200x200km, so this assessment of storm tracks plays a more important role in evaluating storm surge, primarily influenced by the area of low pressure at the centre of the cyclone.

Comparing storm tracks (Figure 7) shows that for 8 of 12 cyclones, the RAL2 storm tracks have at least one ensemble member that makes landfall with the bounds of an IBTrACS track. Notable exceptions to this are: BOB07, which shows high con-
225 sistency in storm track amongst the RAL2 ensemble, but makes landfall to the north of the IBTrACS estimates; TC01B and Viyaru, which show greater spread amongst the RAL2 ensemble members, but consistently make landfall to the south of the IBTrACS estimate. Note that no IBTrACS track data are available for cyclone Fani at the time of writing.

**4.5 Differences between 1.5km and 4.4km model output**

We don't explicitly validate the 1.5km data but summarise differences between the distributions of maximum gust speed and
230 minimum MSLP on a quantile basis, in relation to the 4.4km data (Figure 8). In order to facilitate a fair comparison, we compare identical spatial domains roughly equivalent to the 1.5km model domain (see Figure 2), but with a reduced northern extent to exclude as much mountainous terrain as possible, whilst encompassing the full geographic extent of Bangladesh.

Differences in maximum gust speed footprints, for the 1st to 80th percentiles, of the 1.5km data are order 1 kn faster than the
235 4.4km data. In all cases these differences are sufficiently robust that the 90% highest density interval (HDI) of the differences amongst storms does not overlap zero ([0.3, 1.7] kn; [0.14, 0.86] m/s). For the very highest gust speeds (90th, 95th and 99th percentiles of the 1.5km data) the differences with the 4.4km data shows much greater variability. The 90% HDI does overlap zero, with extremes of the quantile differences ranging from -2.4 kn to 1 kn ([-1.22, 0.50] m/s). Compared to lower percentiles, there are comparatively less data in the extreme upper percentiles, so the large range in this case is expected. Given the
240 relatively robust speed increase seen in the 1.5km data, compared to the 4.4km data, for lower percentiles, we suspect that the minimal difference seen in the upper extreme percentiles results from under sampling rather than a systematic difference. Although we might expect the speed increase in the 1.5km data to be consistent across all percentiles given better sampling, we cannot draw this conclusion based on these 12 storms alone.

245 For minimum MSLP footprints, the 1st and 5th percentiles of the 1.5km data are [50, 87] hPa and [10, 37] hPa shallower respectively (90% HDI), but note that the equivalent under sampling observed for high percentiles of gust speeds is likely to

be prevalent in the low percentiles of MSLP. All other percentiles do not show any robust differences – the 90% HDI ranges [-11, 12] hPa. We do not feel these results show robust evidence for a systematic difference in MSLP between the 1.5km and 4.4km data.

250

The percentile differences suggest that the environmental MSLP (i.e. high percentiles) on the edge of the cyclone are similar in both the 1.5km and 4.4km simulations. Given the relationship between central pressure deficit (i.e. the difference between the tropical cyclone central pressure and the environmental pressure outside the tropical cyclone), peak wind speed and tropical cyclone size (e.g. Chavas et al., 2017), this comparisons suggests that 1.5km storms may also be smaller in size than the 4.4km

255 storms. This result is commonly cited in analyses of general circulation models (e.g. Bengtsson et al., 1995; Reed and Chavas, 2015; Shaevitz et al., 2014) and reanalysis data (e.g. Malakar et al., 2020; Schenkel and Hart, 2012).

In general, the substantial increase in computing effort required for the 1.5km simulations, over and above the 4.4km simulations, is probably not merited for most applications given the nature of the parametrisation (see discussion in Section 3.3).

260 **4.6 Other notable results**

There is a semi-diurnal sea level pressure oscillation which occurs in the days preceding the minimum in MSLP. This oscillation is particularly noticeable in the ERA5 dataset for storms Aila, Bulbul, Rashmi, Roanu, Sidr and Viyaru, and to a lesser extent in RAL2 cyclones Akash, Mora, Rashmi, Roanu and TC01B (see Appendix A). The IBTrACS data does not capture this oscillation, probably due to the limited time sampling. This may be a manifestation of the diurnal radiation cycle as noted

265 by Tang & Zhang (2016), Dunion et al. (2014, 2019) and Knaff et al. (2019), amongst others. From simulation studies, Tang & Zhang (2016) in particular note that the absence of a diurnal cycle (principally night time cooling) fails to trigger convection outside the cyclone inner core. Night-time cooling and associated destabilization typically enhance the primary storm vortex, eventually promoting the development of outer rain bands and increasing the size of the storm. Where this process is not evident in model simulations, it could diagnose simulations that have not correctly simulated the cyclogenesis stage and are

270 therefore likely to underestimate cyclone intensity. In our case, most RAL2 simulations, as shown in Appendix B, do not start the cyclone simulations sufficiently in advance of the cyclone landfall (for computational efficiency reasons) and we have trimmed the spin-up period from the plots. This means that we cannot fully utilise this observation. Assessment of future tropical cyclone simulations could benefit from earlier initialisation times to investigate this further.

275 It is worth emphasising that the RAL2 model wind speed typically compare more favourably with IBTrACS wind speed data than to ERA5 wind speed. Based on the evaluation of these 12 events, tropical cyclone hazards in the ERA5 deterministic output may underestimated wind and gust intensity, and MSLP depth for tropical cyclones. For some specific cases, despite the ERA5 representation of Fani and Bulbul being less intense compared to the IBTrACS estimates, our RAL2 ensemble has sufficient model freedom (over a 24 hour spin-up period) to develop the ERA5 initial conditions into peak gust and minimum

280  MSLP intensities that have much greater agreement with the IBTrACS data than the ERA5 data. This adds credibility to the spread of the RAL2 model ensembles: where there is substantial RAL2 ensemble spread (e.g. Viyaru or Mora) we suggest this reflects greater atmospheric variability associated with these events, such that the RAL2 ensemble might producing a wider range of counterfactual storm outcomes than would otherwise be seen in the driving reanalysis. Comparing these event ensembles with the ERA5 ensemble spread would be an interesting avenue of future work.

285

**5 Data Access**

RAL2 model (Steptoe et al., 2020) output in NetCDF format is available from https://doi.org/10.5281/zenodo.3600201. All data is licenced under Creative Common Attribution 4.0 International (CC BY 4.0). ERA5 data is available from the Copernicus Climate Change Service portal https://climate.copernicus.eu/climate-reanalysis. IBTrACS version 4 data is available

290  from https://www.ncdc.noaa.gov/ibtracs/index.php?name=ib-v4-access.

**5.1 Compatibility with Oasis Loss Model Framework**

To facilitate integration with loss modelling processing necessary for risk management and risk transfer, we also make data available in a format compatible with the open source Oasis loss model (OASIS LMF, 2020). This data format is designed to be used as one component of a loss model and is formed of CSV and binary files. This data is available under CC-BY 4.0

295  licence from https://oasishub.co/dataset/bangladesh-tropical-cyclone-historical-catalogue.

**6 Conclusions**

To our knowledge, these are the first kilometre scale simulations of tropical cyclones over Bangladesh, using ERA5 data as initial and boundary conditions. We summarise key results as follows:

- RAL2 model ensembles typically compare more favourably with IBTrACS data than the ERA5 data. In general, the
300  RAL2 downscaled wind speeds tend to better capture the amplitude of wind speed increase displayed by the IBTrACS data, than ERA5. This implies tropical cyclone hazards in the ERA5 deterministic output may be underestimated.
- RAL2 model ensemble shows a typical increase in peak gust speed of 41 to 118 knots (relative to ERA5 only) and a deepening in minimum MSLP of up to -27 hPa (relative to ERA5 and IBTrACS).
- Generally, there is greater delay in RAL2 MSLP minima, relative to ERA5 and IBTrACS, than in RAL2 gust speed
305  maxima.
- Cyclones that compare particularly well are Mora (timing and intensity of gust and MSLP) and Aila (track, timing of gust and MSLP).

- Cyclones in the 1.5km dataset have similar MSLP minima, but slightly faster maximum gust speeds. This implies that that tropical cyclones in the 1.5km RAL2 simulations may be smaller in size than the 4.4km tropical cyclones.

310
- Further work comparing the spread of RAL2 ensembles with the ERA5 uncertainty information would contextualise the range of variability that is introduced by the RAL2 model ensemble configuration.

- Further work is needed to identify landfall times based on the RAL2 tracks. Future downscaling simulations would benefit from outputting variables required for tracking at hourly intervals, to facilitate hourly storm tracking.

**Appendix A Supplementary Data Descriptions**

315 **A1 RAL2 Time Methods**

Time methods are defined by the sampling period of the data and the sampling type applied to this period. The sampling period (or sampling interval) is one of: hourly (T1H), 3-hourly (T3H) or 24-hourly (T24H). The sampling type is one of max (maximum), min (minimum), mean or point. Point sampling is an instantaneous sample taken from the model time-step (which is typically much less than the sample period). Together then, T1Hmax is interpreted as hourly maximum data; T3Hmean is

320 interpreted a 3-hourly mean data, and T1Hpoint are model instantaneous time-step output taken every hour.

In addition to timeseries data, we produce time-aggregated data for each ensemble member. Referred to as event 'footprints', variables are aggregated by minima or maxima over the entire time period. These are commonly used within the catastrophe modelling industry.

325 **A2 RAL2 File naming**

Model time-series files are named according to the following convention:

VAR.TIMEMETHOD.UMRA2T.TIMEPERIOD.NAME.RES.nc

330 where: VAR is a short variable identifier of the variable contained within the netCDF file; TIMEMETHOD is the time method, specifying if the var is a mean, min, max or point and the period of time over which the mean, min, max or point measure is found (as described above); UMRA2T is an identifier for the Met Office regional model type; TIMEPERIOD is the time period that the data spans, in the form START_END formatted as YYYYMMDD; NAME is the common name of the storm for the given time period; RES is the resolution of the dataset, either 4p4km = 4.4km or 1p5km = 1.5km grid size.

335

Files relating to ensemble footprints have a simpler file naming structure: fpens.VAR.TIMEMETHOD.NAME.RES.nc

**Appendix B Additional Validation Figures**

**Code availability**

340    The Met Office Unified Model is available for use under licence. A number of research organisations and national meteoro-logical services use the UM in collaboration with the Met Office to undertake basic atmospheric process re- search, produce forecasts, develop the UM code, and build and evaluate Earth system models. For further information on how to apply for a licence, see http://www.metoffice.gov.uk/research/ modelling-systems/unified-model

345    Python and R code used to process the RAL2 data is available at https://doi.org/10.5281/zenodo.3953773

**Author contribution**

HS prepared the manuscript with contributions from all authors.  SW led development of the RAL2 model and SS, NS and HS configured the regional Bangladesh domain and ran the downscaling simulations.  NS & ZM performed the storm tracking. HS, KS and ZM performed the data analysis.

350    **Competing Interests**

The authors declare that they have no competing interests.

**Acknowledgements**

This study is part of the Oasis Platform for Climate and Catastrophe Risk Assessment – Asia (https://www.international-climate-initiative.com/en/nc/details/project/oasis-platform-for-climate-and-catastrophe-risk-assessment-asia-18_II_165-

355    3018), a project funded by the International Climate Initiative (IKI), supported by the Federal Ministry for the Environment, Nature Conservation and Nuclear Safety, based on a decision of the German Bundestag.

The authors would like to thank Adrian Champion, Richard Jones & Jane Strachan for comments that improved early versions of this manuscript.

360    **References**

Bechtold, P. and Bidlot, J.-R.: Parametrization of convective gusts, ECMWF Newsl., (119), 15–18, doi:10.21957/kfr42kfp8c,

2009.

Bengtsson, L., Botzet, M. and Esch, M.: Hurricane-type vortices in a general circulation model, Tellus A Dyn. Meteorol. Oceanogr., 47(2), 175–196, doi:10.3402/tellusa.v47i2.11500, 1995.

365   Bonanno, R., Lacavalla, M. and Sperati, S.: A new high-resolution Meteorological Reanalysis Italian Dataset: MERIDA, Q. J. R. Meteorol. Soc., 145(721), 1756–1779, doi:10.1002/qj.3530, 2019.

Brown, A., Milton, S., Cullen, M., Golding, B., Mitchell, J. and Shelly, A.: Unified Modeling and Prediction of Weather and Climate: A 25-Year Journey, Bull. Am. Meteorol. Soc., 93(12), 1865–1877, doi:10.1175/BAMS-D-12-00018.1, 2012.

Bush, M., Allen, T., Bain, C., Boutle, I., Edwards, J., Finnenkoetter, A., Franklin, C., Hanley, K., Lean, H., Lock, A., Manners, 370   J., Mittermaier, M., Morcrette, C., North, R., Petch, J., Short, C., Vosper, S., Walters, D., Webster, S., Weeks, M., Wilkinson, J., Wood, N. and Zerroukat, M.: The first Met Office Unified Model–JULES Regional Atmosphere and Land configuration, RAL1, Geosci. Model Dev., 13(4), 1999–2029, doi:10.5194/gmd-13-1999-2020, 2020.

C3S: ERA5: Fifth generation of ECMWF atmospheric reanalyses of the global climate, [online] Available from: https://cds.climate.copernicus.eu/cdsapp#!/home, 2017.

375   Chavas, D. R., Reed, K. A. and Knaff, J. A.: Physical understanding of the tropical cyclone wind-pressure relationship, Nat. Commun., 8(1), 1360, doi:10.1038/s41467-017-01546-9, 2017.

Dunion, J. P., Thorncroft, C. D. and Velden, C. S.: The Tropical Cyclone Diurnal Cycle of Mature Hurricanes, Mon. Weather Rev., 142(10), 3900–3919, doi:10.1175/MWR-D-13-00191.1, 2014.

Dunion, J. P., Thorncroft, C. D. and Nolan, D. S.: Tropical Cyclone Diurnal Cycle Signals in a Hurricane Nature Run, Mon. 380   Weather Rev., 147(1), 363–388, doi:10.1175/MWR-D-18-0130.1, 2019.

Harrell, F. and Davis, C. E.: A new distribution-free quantile estimator, Biometrika, 69(3), 635–640, doi:10.1093/biomet/69.3.635, 1982.

Hersbach, H., De Rosnay, P., Bell, B., Schepers, D., Simmons, A., Soci, C., Abdalla, S., Balmaseda, A., Balsamo, G., Bechtold, P., Berrisford, P., Bidlot, J., De Boisséson, E., Bonavita, M., Browne, P., Buizza, R., Dahlgren, P., Dee, D., Dragani, R., 385   Diamantakis, M., Flemming, J., Forbes, R., Geer, A., Haiden, T., Hólm, E., Haimberger, L., Hogan, R., Horányi, A., Janisková, M., Laloyaux, P., Lopez, P., Muñoz-Sabater, J., Peubey, C., Radu, R., Richardson, D., Thépaut, J.-N., Vitart, F., Yang, X., Zsótér, E. and Zuo, H.: Operational global reanalysis: progress, future directions and synergies with NWP including updates on the ERA5 production status, ERA Rep. Ser. No. 27, doi:10.21957/tkic6g3wm, 2018.

IBTrACS: International Best Track Archive for Climate Stewardship (IBTrACS). Technical Documentation, Natl. Ocean. Atmos. Adm. Natl. Clim. Data Cent., 1–24 [online] Available from: https://www.ncdc.noaa.gov/ibtracs/pdf/IBTrACS_version4_Technical_Details.pdf%0Ahttps://www.ncdc.noaa.gov/ibtracs/index.php, 2019.

Kaur, M., Krishna, R. P. M., Joseph, S., Dey, A., Mandal, R., Chattopadhyay, R., Sahai, A. K., Mukhopadhyay, P. and Abhilash, S.: Dynamical downscaling of a multimodel ensemble prediction system: Application to tropical cyclones, Atmos. Sci. Lett., n/a(n/a), doi:10.1002/asl.971, 2020.

Knaff, J. A., Slocum, C. J. and Musgrave, K. D.: Quantification and Exploration of Diurnal Oscillations in Tropical Cyclones, Mon. Weather Rev., 147(6), 2105–2121, doi:10.1175/MWR-D-18-0379.1, 2019.

Knapp, K. R., Kruk, M. C., Levinson, D. H., Diamond, H. J. and Neumann, C. J.: The International Best Track Archive for Climate Stewardship (IBTrACS), Bull. Am. Meteorol. Soc., 91(3), 363–376, doi:10.1175/2009BAMS2755.1, 2010.

Knapp, K. R., Diamond, H. J., Kossin, J. P., Kruk, M. C. and Schreck, C. J. I.: International Best Track Archive for Climate Stewardship (IBTrACS) Project, Version 4, NOAA National Centers for Environmental Information., 2018.

Kruk, M. C., Knapp, K. R. and Levinson, D. H.: A Technique for Combining Global Tropical Cyclone Best Track Data, J. Atmos. Ocean. Technol., 27(4), 680–692, doi:10.1175/2009JTECHA1267.1, 2010.

Leutwyler, D., Lüthi, D., Ban, N., Fuhrer, O. and Schär, C.: Evaluation of the convection-resolving climate modeling approach on continental scales, J. Geophys. Res. Atmos., 122(10), 5237–5258, doi:10.1002/2016JD026013, 2017.

Lock, A., Edwards, J. and Boutle, I.: Unified Model Documentation Paper 024: The Parametrization of Boundary Layer Processes., 2019.

Mahala, B. K., Mohanty, P. K., Das, M. and Routray, A.: Performance assessment of WRF model in simulating the very severe cyclonic storm "TITLI" in the Bay of Bengal: A case study, Dyn. Atmos. Ocean., 88, 101106, doi:10.1016/j.dynatmoce.2019.101106, 2019.

Malakar, P., Kesarkar, A. P., Bhate, J. N., Singh, V. and Deshamukhya, A.: Comparison of Reanalysis Data Sets to Comprehend the Evolution of Tropical Cyclones Over North Indian Ocean, Earth Sp. Sci., 7(2), e2019EA000978, doi:10.1029/2019EA000978, 2020.

OASIS LMF: Oasis Loss Modelling Framework v1.7.0, [online] Available from: https://github.com/OasisLMF/OasisLMF/releases/tag/1.7.0, 2020.

Owens, R. G. and Hewson, T. D.: ECMWF Forecast User Guide, ECMWF, Reading, UK., 2018.

Reed, K. A. and Chavas, D. R.: Uniformly rotating global radiative-convective equilibrium in the Community Atmosphere Model, version 5, J. Adv. Model. Earth Syst., 7(4), 1938–1955, doi:10.1002/2015MS000519, 2015.

Rousselet, G. A. and Wilcox, R. R.: Reaction times and other skewed distributions:problems with the mean and the median, PsyArXiv, doi:10.31234/osf.io/3y54r, 2019.

Rousselet, G. A., Pernet, C. R. and Wilcox, R. R.: Beyond differences in means: robust graphical methods to compare two groups in neuroscience, Eur. J. Neurosci., 46(2), 1738–1748, doi:10.1111/ejn.13610, 2017.

Schenkel, B. A. and Hart, R. E.: An Examination of Tropical Cyclone Position, Intensity, and Intensity Life Cycle within Atmospheric Reanalysis Datasets, J. Clim., 25(10), 3453–3475, doi:10.1175/2011JCLI4208.1, 2012.

Shaevitz, D. A., Camargo, S. J., Sobel, A. H., Jonas, J. A., Kim, D., Kumar, A., LaRow, T. E., Lim, Y.-K., Murakami, H., Reed, K. A., Roberts, M. J., Scoccimarro, E., Vidale, P. L., Wang, H., Wehner, M. F., Zhao, M. and Henderson, N.: Characteristics of tropical cyclones in high-resolution models in the present climate, J. Adv. Model. Earth Syst., 6(4), 1154–1172, doi:10.1002/2014MS000372, 2014.

Singh, K. S. and Bhaskaran, P. K.: Prediction of landfalling Bay of Bengal cyclones during 2013 using the high resolution Weather Research and Forecasting model, Meteorol. Appl., 27(1), e1850, doi:10.1002/met.1850, 2020.

Skamarock, W. C., Klemp, J. B., Dudhia, J., Gill, D. O., Liu, Z., Berner, J., Wang, W., Powers, J. G., Duda, M. G., Barker, D. M. and Huang, X.-Y.: A Description of the Advanced Research WRF Version 4, , 145, doi:10.5065/D6MK6B4K, 2019.

Srinivas, C. V, Bhaskar Rao, D. V, Yesubabu, V., Baskaran, R. and Venkatraman, B.: Tropical cyclone predictions over the Bay of Bengal using the high-resolution Advanced Research Weather Research and Forecasting (ARW) model, Q. J. R. Meteorol. Soc., 139(676), 1810–1825, doi:10.1002/qj.2064, 2013.

Steptoe, H., Savage, N., Sadri, S., Salmon, K., Maalick, Z. and Webster, S.: Bangladesh - Tropical Cyclone Historical Catalogue, Zenodo, (data set), doi:10.5281/zenodo.3600201, 2020.

Taddei, S., Capecchi, V., Pasi, F., Vannucchi, V. and Bendoni, M.: Downscaling ERA-5 reanalysis data for coastal short-term and long-term risk assessment in the North Western Mediterranean sea ., , 21, 18262, 2019.

Tang, X. and Zhang, F.: Impacts of the Diurnal Radiation Cycle on the Formation, Intensity, and Structure of Hurricane Edouard (2014), J. Atmos. Sci., 73(7), 2871–2892, doi:10.1175/JAS-D-15-0283.1, 2016.

Tiedtke, M.: A Comprehensive Mass Flux Scheme for Cumulus Parameterization in Large-Scale Models, Mon. Weather Rev., 117(8), 1779–1800, doi:10.1175/1520-0493(1989)117<1779:ACMFSF>2.0.CO;2, 1989.

Ullrich, P. A. and Zarzycki, C. M.: TempestExtremes: a framework for scale-insensitive pointwise feature tracking on unstructured grids, Geosci. Model Dev., 10(3), 1069–1090, doi:10.5194/gmd-10-1069-2017, 2017.

Wang, X., Tolksdorf, V., Otto, M. and Scherer, D.: WRF-based dynamical downscaling of ERA5 reanalysis data for High Mountain Asia: Towards a new version of the High Asia Refined analysis, Int. J. Climatol., joc.6686, doi:10.1002/joc.6686, 2020.

Wilcox, R. R. and Erceg-Hurn, D. M.: Comparing two dependent groups via quantiles, J. Appl. Stat., 39(12), 2655–2664, doi:10.1080/02664763.2012.724665, 2012.

Wilcox, R. R., Erceg-Hurn, D. M., Clark, F. and Carlson, M.: Comparing two independent groups via the lower and upper quantiles, J. Stat. Comput. Simul., 84(7), 1543–1551, doi:10.1080/00949655.2012.754026, 2014.

[Figure]

**Figure 1** Ensemble configuration for the RAL2-C (UM) downscaling suite. ERA5 initial conditions (orange dots) initialise the simulation start point (green dots). Each ensemble member then has a 24 hour spin-up period (grey dashed lines) which is discarded from all analysis. The 48-hour simulation that is kept is represented by the solid blue line. ERA5 lateral boundary conditions (LBCs, black dots) feed into the 4.4km domain every hour. The lagged ensemble is designed to simulate a central 24-hour period (shaded grey), common to all ensemble members and centred on the tropical cyclone land-fall time (orange star), but also sample a range of ERA5 initial conditions.

[Figure]

455

**Figure 1** Model domains used for the 4.4km (red) and 1.5km (blue) regional models. ERA5 data, with global coverage, provides initial conditions for the 4.4km domain. The 1.5km model takes its initial and boundary conditions from the 4.4km model. The domain data mask used for validation plots on Section 3 is in green.

[Figure]

**Figure 2 S**torm specific comparison of maximum gust speed (top), maximum wind speed (middle) and minimum sea-level pressure (bottom) for tropical cyclone Sidr (Nov 2007). The dynamically downscaled, high-resolution Met Office model (RAL2) is shown by the coloured lines, where each individual line represents one ensemble member, where the initialisation time is coloured lighter to darker. These are shown against IBTrACS (grey triangles with uncertainty ranges) and ERA5 (black line). Equivalent plots for other events can be found in Appendix B.

[Figure]

465

**Figure 4** Differences in maximum wind speed intensity (left) and timing of maximum (right) for IBTrACS US (blue) ND (orange) and ERA5 (green) relative to RAL2 ensemble members, ordered by magnitude of the intensity difference. Comparisons are made only within the period of RAL2 data, up to 36 hours pre and post landfall. Differences are calculated relative to RAL2 maximum, such that a positive
470 intensity (time) difference indicates that the RAL2 model is faster (ahead) of the respective ERA5 or IBTrACS data. IBTrACS data is resampled by forward padding data to hourly intervals to aid comparison with RAL2. Where there are joint maxima in the IBTrACS data over multiple timesteps, we plot the smallest differences. Individual model differences are shown by coloured circles, with median difference per storm are show by coloured bars. Lower boxplots aggregate differences across all storms, with the 50th percentile marked by the black bar and whiskers extending to the 5th and 9th percentiles of the data

[Figure]

**Figure 5** Differences in minimum MSLP (left) and time of minimum (right) for IBTrACS US (blue) ND (orange) and ERA5 (green) relative to RAL2 ensemble members, ordered by magnitude of the MSLP intensity difference. Details as for Figure 4. A negative (positive) difference in MSLP indicates that the RAL2 MSLP minima are deeper (shallower) than the respective ERA5 or IBTrACS data. Note that IBTrACS ND MSLP data was not available for Fani or BulBul, and US MSLP not available for BOB01, BOB07 and TC01B, at the time of writing.

475

480

[Figure]

**Figure 6** Differences in maximum 3-second gust speed (left) and timing of maximum gust speed (right) for ERA5 relative to RAL2 ensemble members, ordered by the magnitude of the gust speed difference. Details as for Figure 4. A positive (negative) difference in gust speed indicates that the RAL2 gust speed maximum is faster (slower) than ERA5 data. Note that at the time of writing gust speed data was not available from IBTrACS for any of these events.

485

[Figure]

**Figure 7** Storm track comparisons for IBTrACS US (blue lines) and ND (orange) with RAL2 ensemble track bin densities. Note that IBTrACS data for the most recent cyclone Fani and Bulbul are incomplete at the time of writing. Dashed lines represent variable IBTrACS storm track uncertainty, based on cyclone intensity.

490

[Figure]

**Figure 8** Percentile differences between 1.5km and 4.4km tropical cyclone data for (a) maximum gust speed and (b) minimum mean sea-level pressure (MSLP) footprints. Differences between resolutions are assessed on a quantile basis, in a hierarchical manner to account for dependence between storm ensemble members sampled from multiple storms. Quantile median estimates are shown by black circles, with 95% highest density intervals (HDI) shown by black bars. Where the 95% HDI overlaps 0, the median circles are filled white. The bootstrapped difference distribution (n=1000) at each quantile is shaded turquoise (gust speeds) and orange (MSLP).

500

| Variable | Identifier | Unit |
|---|---|---|
| net down surface sw flux corrected | rsnds | $W\ m^{-2}$ |
| wet bulb potential temperature | wbpt | K |
| air pressure at sea level | psl | Pa |
| air temperature | tas | K |
| geopotential height | zg | M |
| relative humidity | hur | % |
| stratiform rainfall amount | prlst | $kg\ m^{-2}$ |
| stratiform snowfall amount | prlssn | $kg\ m^{-2}$ |
| surface downwelling shortwave flux in air | rsds | $W\ m^{-2}$ |
| wind speed of gust | fg | $m\ s^{-1}$ |
| x wind | ua | $m\ s^{-1}$ |
| y wind | va | $m\ s^{-1}$ |

505

**Table 1** Available model output and their SI units.

| Name | Landfall Date (DD/MM/YYYY HH:MMZ) | IBTrACS ID |
|---|---|---|
| **BOB01** | 30/04/1991 00:00Z | 1991113N10091 |
| **BOB07** | 25/11/1995 09:00Z | 1995323N05097 |
| **TC01B** | 19/05/1997 15:00Z | 1997133N03092 |
| **Akash** | 14/05/2007 18:00Z | 2007133N15091 |
| **Sidr** | 15/11/2007 18:00Z | 2007314N10093 |
| **Rashmi** | 26/10/2008 21:00Z | 2008298N16085 |
| **Aila** | 25/05/2009 06:00Z | 2009143N17089 |
| **Viyaru** | 16/05/2013 09:00Z | 2013130N04093 |
| **Roanu** | 21/05/2016 12:00Z | 2016138N10081 |
| **Mora** | 30/05/2017 03:00Z | 2017147N14087 |
| **Fani** | 04/05/2019 06:00Z | 2019117N05088 |
| **Bulbul** | 09/11/2019 18:00Z | 2019312N16088 |

510  **Table 2** List of tropical cyclones downscaled in this dataset. IBTrACS ID refers to the International Best Track Archive for Climate Stewardship storm identifier. Landfall dates are provided for reference and do not necessarily reflect the landfall date of the downscaled data. Similarly, names are provided as a shorthand identifier, and are used for file naming purposes, but do not necessarily reflect the official storm identifier.

515

| Dataset | Data Type | Spatial Resolution | Temporal Resolution | Compared Variables | Convective/parameter-ised wind speed |
|---|---|---|---|---|---|
| **Downscaled (RAL2) model data** | Gridded | 4.4km | 1-hourly | Gust, Wind, MSLP | Convective permitting |
| **Downscaled (RAL2) model data** | Gridded | 1.5km | 1-hourly | Gust, Wind, MSLP | Convective permitting |
| **ERA5** | Gridded | 30km | 1-hourly | Gust, Wind, MSLP | Parameterised |
| **IBTrACS v4, US** | Time Series | 10km (0.1°) | 3-hourly | Wind, MSLP | Observed from various sources |
| **IBTrACS v4, India** | Time Series | 10km (0.1°) | Interpolated to 3-hourly (most data reported at 6 hourly) | Wind, MSLP | Observed from various sources |

**Table 3** Datasets and their key characteristics used in the model validation.

520

[Figure]

**Figure B1** Comparison of maximum wind/gust speed and minimum sea-level pressure for tropical cyclone Aila (May 2009). Details as for Figure 3.

[Figure]

**Figure B2** Comparison of maximum wind/gust speed and minimum sea-level pressure for tropical cyclone Akash (May 2007). Details as for Figure 3.

[Figure]

**Figure B3** Comparison of maximum wind/gust speed and minimum sea-level pressure for tropical cyclone BOB01 (Apr 1991). Details as for Figure 3.

[Figure]

**Figure B4** Comparison of maximum wind/gust speed and minimum sea-level pressure for tropical cyclone BOB07 (Nov 1995). Details as for Figure 3.

[Figure]

**Figure B5** Comparison of maximum wind/gust speed and minimum sea-level pressure for tropical cyclone Bulbul (Nov 2019). Details as for Figure 3.

[Figure]

**Figure B6** Comparison of maximum wind/gust speed and minimum sea-level pressure for tropical cyclone Fani (May 2009). Details as for Figure 3.

[Figure]

**Figure B7** Comparison of maximum wind/gust speed and minimum sea-level pressure for tropical cyclone Mora (May 2017). Details as for Figure 3.

[Figure]

**Figure B8** Comparison of maximum wind/gust speed and minimum sea-level pressure for tropical cyclone Rashmi (Oct 2008). Details as for Figure 3.

[Figure]

**Figure B9** Comparison of maximum wind/gust speed and minimum sea-level pressure for tropical cyclone Roanu (May 2016). Details as for Figure 3.

[Figure]

**Figure B10** Comparison of maximum wind/gust speed and minimum sea-level pressure for tropical cyclone TC01B (May 1997). Details as for Figure 3.

[Figure]

**Figure B11** Comparison of maximum wind/gust speed and minimum sea-level pressure for tropical cyclone Viyaru (May 2013). Details as for Figure 3.

---

## Author Response (AR2)

**Reply to Report #1: Review of nhess-2020-299**

We thank the reviewer for their comments. Our replies are inline below:

**Specific comments**

**I have a comment on the response and paper edits from Reviewer 2 comment 3 about wind speed discrepancy.**

**Manuscript extract:**

**"In general, median peak gust speeds from the RAL2 model ensemble are found to be 22 to 43 m s-1 faster compared to ERA5, but it is known that extreme gusts associated with vigorous convection in ERA5 are generally under-estimated, sometimes by a factor of two (Owens and Hewson, 2018)."**

**My two questions are:**

**1. Can you add another comment about the implication of this bias? Are you suggesting that the bias is only in ERA5 and the RAL2 winds are correct, or might they also have errors associated with them.**

> Generally, only grid spacings on the order of 1 km are comparable to the size of particularly energetic eddies in the planetary boundary layer (e.g. Leutwyler et al., 2017), so the turbulent processes as well as the dominant turbulent length scale will be under resolved in both our downscaled model and ERA5. We have added text to Section 2 to clarify this.

**2. Can you comment on the difference between IBTrACS and RAL2? If they are more similar than RAL2 and ERA5, this would help to justify ERA5 errors in wind gusts. Originally the statement was that this 22-43 m/s error was in ERA5 and IBTrACS, but was this incorrect?**

> These numbers refer to the gust speed difference against ERA5 only. It was incorrect to reference IBTrACS in association with these numbers, as IBTrACS data does not estimate gust speed. The median difference for wind speed (across all events) is 18 m s$^{-1}$ for ERA5. Comparing IBTrACS wind speeds, median difference against the US forecast is $-3$ m s$^{-1}$ and against the New Delhi forecast is 5m s$^{-1}$. We have added a sentence in Section 2 to clarify this.

**Line 82: Is the model called RA2 or RAL2?**

> Thank you for highlighting this. For consistency we use 'RAL2'.

**Line 197: Missing 'to'**

> Corrected.

**Reply to Report #2: Review of nhess-2020-299**

We thank the reviewer for their comments. Our replies are inline below:

**Please explain in greater detail what value the hazard maps here would add to the short-term forecast problem in the situation where the same model could be run in ensemble NWP mode in real time using appropriate initial conditions. Why is that by itself not the best way to produce forecast warnings?**

> The focus of this paper is not about short-term forecasting. As you describe, ensemble NWP (and implicitly processes such as data assimilation) are the best solution for forecasting a specific event at lead-times of hours to days. The process we describe in Section 3.1 could just as well be applied to ensemble NWP data, as our hazard maps. We have added words in Section 3.1 to try and clarify this.

**The paper really needs to have some more explanation of key aspects of the underlying simulations: domain size, duration of simulations and lead time before landfall, a little information about physical parameterizations, definition of a gust, how the simulations were validated, etc. This information is really essential and the reader shouldn't be made to look up a different paper for it (particularly one that isn't published yet, and might not be, as far as we know). The present paper is short and could easily include a new section on this.**

> We have added the following to Section 2:

Each ensemble member requires a 24-hour spin-up period as the RAL2 model adjusts from weak initial conditions taken from the ERA5 driving global model (of Hersbach et al., 2020). This initial 24 hours of model data are discarded in subsequent analysis and data files. Thereafter, each ensemble member is free running for a further 48 hours, with hourly boundary conditions provided by ERA5. Collectively, the ensembles members sample a range of lead times before landfall from 12-36 hours.

The parameterised RAL2 gust diagnostic represents a prediction of the 3-second average windspeed at every timestep. The maximum of this 3-second average speed over an hour is then taken to give the hourly maximum 3-second gust speed. While not truly resolving deep convection, RAL2 is able to explicitly represent deep convective processes within the resolved dynamics. At these kilometre-scale resolutions the lower horizontal size limit of convective cells is still set by the effective resolution of 5 to 10 times the grid length (Boutle et al., 2014; Skamarock, 2004). Generally, only grid spacings on the order of 1 km are comparable to the size of particularly energetic eddies in the planetary boundary layer (Leutwyler et al., 2017), so the turbulent processes as well as the dominant turbulent length scale will be under resolved in our downscaled model (and also ERA5). The RAL2 model uses a gust parametrisation based on 10 m wind speed with scaling proportional to the standard deviation of the horizontal wind that also accounts for friction velocity, atmospheric stability and roughness length (Lock et al., 2019).

> Section 2 already includes details of the domain size. The paper describing the modelling set-up and data validation has now been accepted for publication in Scientific Data (Steptoe et al., 2021).

**References**

Boutle, I. A., Eyre, J. E. J., & Lock, A. P. (2014). Seamless Stratocumulus Simulation across the Turbulent Gray Zone. *Monthly Weather Review*, *142*(4), 1655–1668.

https://doi.org/10.1175/MWR-D-13-00229.1

Hersbach, H., Bell, B., Berrisford, P., Hirahara, S., Horányi, A., Muñoz-Sabater, J., et al. (2020). The ERA5 global reanalysis. *Quarterly Journal of the Royal Meteorological Society*, *146*(730), 1999–2049. https://doi.org/https://doi.org/10.1002/qj.3803

Leutwyler, D., Lüthi, D., Ban, N., Fuhrer, O., & Schär, C. (2017). Evaluation of the convection-resolving climate modeling approach on continental scales. *Journal of Geophysical Research: Atmospheres*, *122*(10), 5237–5258. https://doi.org/https://doi.org/10.1002/2016JD026013

Lock, A., Edwards, J., & Boutle, I. (2019). *Unified Model Documentation Paper 024: The Parametrization of Boundary Layer Processes*.

Skamarock, W. C. (2004). Evaluating Mesoscale NWP Models Using Kinetic Energy Spectra. *Monthly Weather Review*, *132*(12), 3019–3032. https://doi.org/10.1175/MWR2830.1

Steptoe, H., Savage, N., Sadri, S., Salmon, K., Maalick, Z., & Webster, S. (2021). Tropical cyclone simulations over Bangladesh at convection permiting 4.4km & 1.5km resolution. *Scientific Data*, (accepted). https://doi.org/10.1038/s41597-021-00847-5